# Geometry-Guided Modeling of Foundation Features Enables Generalizable Object Shape Deformation Learning

Yiyao Ma [1]   Kai Chen [1]   Zhongxiang Zhou [2]   Zhuheng Song [1]   Dongsheng Xie [1]
Zelong Tan [1]   Rong Xiong [2 3]   Qi Dou [1]

## Abstract

Monocular 3D shape recovery is fundamental to geometric understanding, yet achieving robust generalization across arbitrary viewpoints and unseen object categories remains a significant challenge. In this paper, we present a generalizable deformation learning framework that reconstructs 3D objects by explicitly deforming a category-level shape template to match the target observation. To address complex shape variations between the template and the target, we introduce a geometry-guided feature modeling mechanism. This process first enriches foundation features with template topology to yield a geometry-aware representation, which is then explicitly correlated with the target observation to guide precise deformation. Furthermore, to bridge the disparity between the fixed template and arbitrary target views, we propose a view-adaptive feature aggregation module. This module leverages multi-view template features and their corresponding camera poses to enrich the canonical template representation, ensuring robust feature alignment regardless of the target's perspective. Extensive experiments demonstrate that our approach significantly outperforms state-of-the-art methods in handling large shape variations and diverse viewpoints, exhibiting strong generalization to novel categories and effectively supporting downstream real-world dexterous robotic manipulation tasks. Project homepage: https://GODeform.github.io/

## 1. Introduction

Recovering object shape from a monocular image is fundamental to geometric understanding and spatial reasoning (Mescheder et al., 2019; Fahim et al., 2021). However, relying on a single viewpoint impedes the reconstruction of plausible geometry for invisible regions, while the vast structural complexity of the real world demands strong generalization capabilities to handle shapes beyond the training distribution (Wang et al., 2024; Jiang et al., 2025). Despite growing attention in recent years (Hong et al., 2024; Long et al., 2024; Wang et al., 2025), robust and generalizable shape recovery across viewpoints and diverse object categories remains a critical challenge (Cho et al., 2025).

Recent advances leverage generative models to achieve high-fidelity reconstruction (Long et al., 2024). However, their performance is often sensitive to viewpoint variations and partial observability. In self-occluded regions, they frequently hallucinate geometry that is unrealistic or inconsistent with visible surfaces (Huang et al., 2025; Wu et al., 2025). While some works further incorporate a category-level template to inject structural priors (Xiu et al., 2022; Zheng et al., 2021; Wang et al., 2025), they typically treat the template merely as a source of feature conditioning rather than a geometric starting point. Without explicitly establishing point-wise relationships to the template, they struggle to effectively leverage the template's topology to regularize the reconstruction, and thus remain prone to structural degradation in invisible parts (Xiu et al., 2023).

To address these structural issues, another line of research focuses on explicitly deforming a template shape to match the target observation (Wang et al., 2018; Wen et al., 2019; Sommer et al., 2025; Shuai et al., 2023). By warping a known geometry, these methods leverage the template's structural priors to ensure plausibility in occluded regions, while inherently preserving the template's topology to maintain dense correspondence (Groueix et al., 2018; Chen et al., 2025; Liu et al., 2023a). However, these approaches typically rely on visual encoders trained from scratch on limited datasets, which struggle to extract robust semantic features for unseen categories (Wallace & Hariharan, 2019). Furthermore, even within seen categories, they lack a deep understanding of

[1]Department of Computer Science and Engineering, The Chinese University of Hong Kong, Hong Kong SAR, China [2]Zhejiang Innovation Center for Humanoid Robotics, Ningbo, China [3]State Key Laboratory of Industrial Control and Technology, Zhejiang University, Hangzhou, China. Correspondence to: Kai Chen <kaichen@cuhk.edu.hk>.

*Proceedings of the 43$^{rd}$ International Conference on Machine Learning*, Seoul, South Korea. PMLR 306, 2026. Copyright 2026 by the author(s).

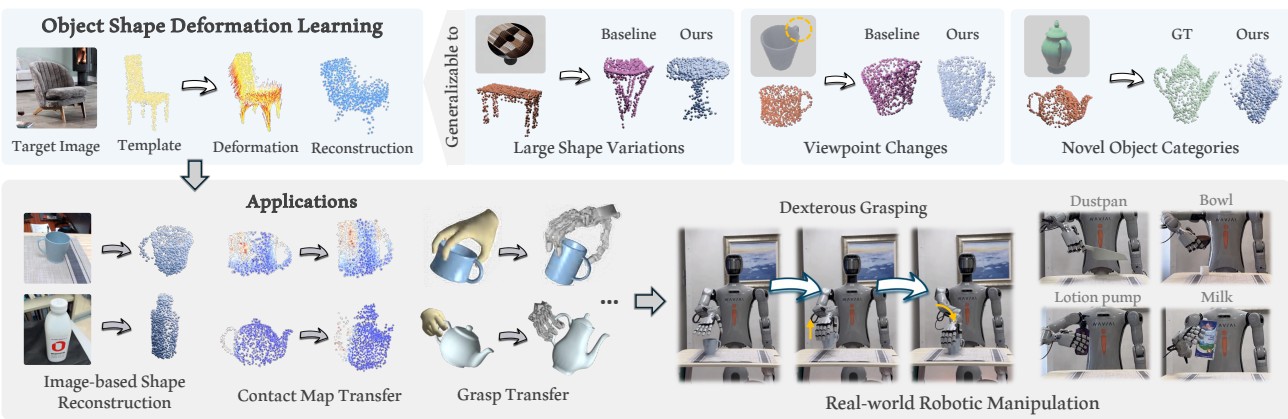

*Figure 1.* The proposed object shape deformation learning framework can handle large template-target shape variations, remains robust to diverse camera viewpoints, and generalizes to unseen categories. It enables various downstream applications, and effectively supports generalizable dexterous robotic manipulation in the real world.

the intrinsic geometric relationship between the template and diverse novel objects, and thus can fail to predict accurate deformation fields when the target shape deviates significantly from the template (Pan et al., 2019; Zhang et al., 2024; Di et al., 2024). These limitations motivate the need for a deformation learning framework that generalizes across viewpoints, large template-target shape variations, and object categories beyond training.

In this paper, we present a generalizable deformation learning framework enabled by geometry-guided modeling of foundation features. Our core insight is to harness the robust semantic similarity provided by 2D foundation models (Siméoni et al., 2025) to guide 3D shape deformation. The pre-trained foundation features offer a consistent representation across categories and domains, strengthening robustness under distribution shifts and supporting generalization to unseen categories. To handle complex shape variations between the template and diverse target objects, we introduce a geometry-guided feature propagation and alignment module. Specifically, we build a geometry-aware template representation by enriching 2D foundation features with template shape features and propagating them over the template surface to cover both visible and occluded regions. We then explicitly model the relationship between the resulting template representation and the target observation to condition the inference of a per-point deformation field. Together, these components enable the model to capture fine-grained similarities between the template and diverse targets, yielding deformation predictions that generalize across viewpoints, shape variations, and object categories.

However, the efficacy of such feature guidance is challenged when there is a significant viewpoint disparity between the fixed template view and the target observation. Such misalignment diminishes the semantic correspondence captured by foundation models, potentially degrading deformation quality. To achieve robustness across arbitrary observation angles, we introduce a view-adaptive feature aggregation module to upgrade the template representation. Instead of relying on a static template view, this module dynamically identifies a primary view from a pre-sampled support set. We then enhance this primary representation by injecting complementary cues from auxiliary template views via a pose-aware attention mechanism that explicitly accounts for relative camera geometry. The resulting fused representation can effectively mitigate perspective-induced feature drift and stabilize deformation prediction even under large viewpoint changes. As illustrated in Figure 1, our proposed deformation learning framework proves robust to large shape variations, viewpoint changes, and novel object categories, thereby facilitating a wide range of downstream applications. Our contributions are summarized as follows:

- We propose a generalizable deformation learning framework that reconstructs 3D objects by explicitly deforming category-level templates, leveraging the robust visual representations of 2D foundation models to generalize to unseen categories.

- We introduce a geometry-guided feature modeling mechanism that enriches foundation features with template topology to establish precise point-wise correspondences, effectively addressing complex shape variations between the template and the target.

- We design a view-adaptive feature aggregation module that dynamically incorporates camera pose embeddings to compensate for perspective-induced geometric misalignments, ensuring robustness against significant viewpoint disparities.

- Extensive experiments show that our method generalizes well across diverse viewpoints, large shape variations, and novel object categories, while effectively facilitating generalizable dexterous robotic manipulation in the real world.

# 2. Related Work

## 2.1. Object Shape Reconstruction via Deformation

A prevalent paradigm in category-level 3D reconstruction involves deforming a canonical shape template to recover the target geometry. Early approaches primarily focused on sparse deformation mechanisms, employing low-dimensional controls such as cages (Yifan et al., 2020), semantic keypoints (Jakab et al., 2021), or volumetric warps (Jack et al., 2018; Kurenkov et al., 2018) to constrain the solution space, while others explored part-based deformation (Shuai et al., 2023; Paschalidou et al., 2021) to handle structural variations. To capture more fine-grained geometric details beyond these sparse controls, subsequent research shifted towards dense deformation strategies, predicting high-dimensional fields like point-wise offsets (Wang et al., 2019) or pixel-aligned displacements (Wang et al., 2018; Wen et al., 2019) to refine local surface variations. To further improve reconstruction fidelity, recent works propose retrieval-augmented pipelines that first retrieve shape templates visually similar to the target from a library, followed by deformation to fit the observed images (Di et al., 2023; Zhang et al., 2024; Di et al., 2024; Uy et al., 2021). However, the performance of these methods is heavily contingent on the geometric similarity between the retrieved template and the target object. Furthermore, these deformation models typically fail to generalize beyond their training categories, limiting their practical usage in open-world scenarios. In contrast, our method explicitly models foundation features to guide the shape deformation learning, thereby enabling superior generalizability across diverse shape variations and unseen object categories.

## 2.2. Object Representation with Foundation Models

Large-scale pre-trained 2D foundation models have demonstrated strong generalization across categories and domains (Radford et al., 2021; Oquab et al., 2024; Siméoni et al., 2025), and their representations have been widely adopted as generic visual descriptors for diverse downstream tasks (Chen et al., 2024b; Barsellotti et al., 2025; Lin et al., 2024). Despite these advances in 2D, extending such capabilities to 3D remains challenging. Although recent efforts explore native 3D foundation models (Pang et al., 2023; Xue et al., 2023; Liu et al., 2023b), their scalability is fundamentally constrained by the scarcity and limited diversity of high-quality 3D data. As a result, they often exhibit limited generalization ability and a constrained understanding of complex semantic concepts compared to their 2D counterparts (Thengane et al., 2025). To bridge this gap, a growing body of work adapts pre-trained 2D foundation models for 3D understanding by lifting and aligning 2D features with geometric representations, achieving promising results in scene understanding (Knaebel et al., 2026; Zhu et al., 2024;

2025; Zhu et al.), pose estimation (Caraffa et al., 2024; Chen et al., 2024a), and robotic manipulation (Huang et al., 2023; Chen et al., 2026). Inspired by these successes, we study generalizable object shape deformation by making 2D foundation features geometry-aware on the template surface, enabling robust point-wise template-target correspondence reasoning.

# 3. Methodology

We formulate the task as a *geometry-guided shape deformation* problem. Given a single RGB image $I_{\mathcal{T}}$ of a target object and a category-level 3D template $\mathcal{S} \in \mathbb{R}^{N \times 3}$, our goal is to predict a per-point deformation field $\mathcal{D} \in \mathbb{R}^{N \times 3}$ that aligns the template geometry with the observed target shape. Unlike standard reconstruction approaches, this formulation yields both the recovered 3D shape and explicit point-wise correspondences between the template and the target. To achieve this, we model the deformation process as a conditional flow matching problem. We define the conditioning context as $\mathbf{c} = \{I_{\mathcal{T}}, I_{\mathcal{S}}\}$, where $I_{\mathcal{S}}$ denotes the associated template image. Our objective is to learn a continuous deformation mapping $\Phi$ that transports the geometric distribution of the template $\mathcal{S}$ to the target $\hat{\mathcal{T}} = \Phi(\mathcal{S} \mid \mathbf{c})$. This ensures that the reconstruction originates from a topologically plausible prior while effectively recovering instance-specific fine details through the learned flow.

## 3.1. Shape Deformation Learning via Flow Matching

### 3.1.1. FLOW MATCHING FORMULATION

Flow matching provides a framework for learning continuous deformation paths from a template $\mathcal{S}$ to a target $\mathcal{T}$ under context $\mathbf{c}$. The flow is governed by an ordinary differential equation (ODE) $d\phi_t/dt = \mathbf{v}_t(\phi_t \mid \mathbf{c})$, transporting the distribution from $p(\mathbf{x} \mid \mathcal{S})$ to $p(\mathbf{x} \mid \mathcal{T})$. Following (Albergo & Vanden-Eijnden, 2023; Liu et al., 2023d), we construct a linear interpolation path $\mathbf{x}_t = (1 - t)\mathbf{x}_0 + t\mathbf{x}_1$ between correspondences, which implies a constant target velocity $\mathbf{u}_t = \mathbf{x}_1 - \mathbf{x}_0$. We train a network $v_\theta(\mathbf{x}_t, t, \mathbf{c})$ to approximate this velocity field. At inference time, we achieve efficiency by fixing $t = 0$ and directly predicting the deformation $\mathcal{D}$:

$$\mathcal{D} = v_\theta(\mathcal{S}, 0, \mathbf{c}), \quad \hat{\mathcal{T}} = \mathcal{S} + \mathcal{D}. \tag{1}$$

This enables single-step inference while maintaining the theoretical guarantees of continuous flow matching.

### 3.1.2. GEOMETRY-GUIDED FOUNDATION FEATURE MODELING

Guiding object shape deformation using 2D images of template and target objects presents a significant challenge, as the conditioning process requires robustness to target varia-

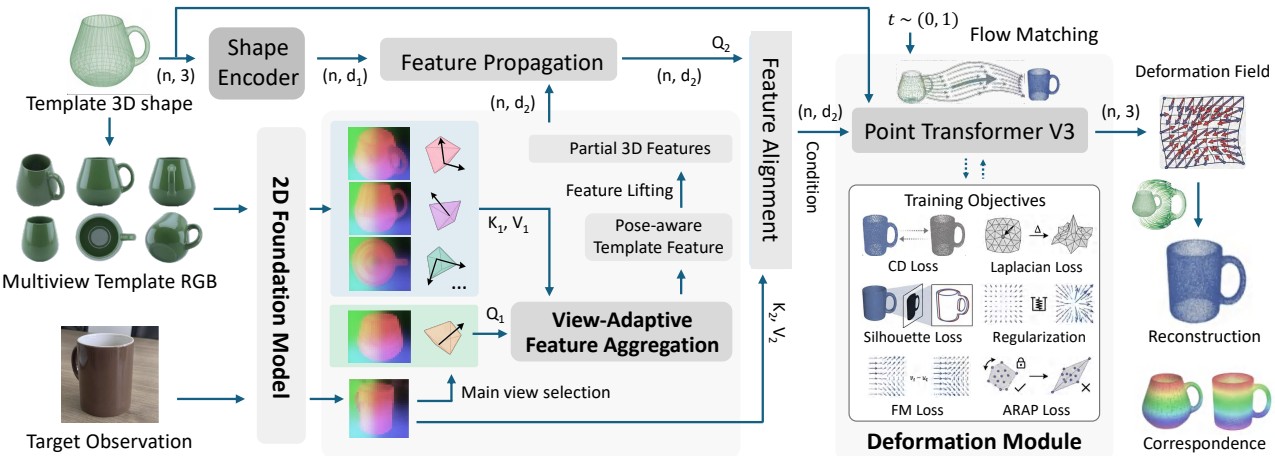

*Figure 2.* Overview of our proposed framework. The core of our approach is a conditional flow-matching module that warps a template shape toward a target via a continuous trajectory. This deformation is conditioned on the geometry-guided modeling of 2D foundation features. To ensure these features are spatially aligned and robust to varying observation angles, we introduce two key components: (1) a geometry-guided feature modeling process, which diffuses lifted 2D features across the 3D template surface to bridge the domain gap; and (2) a view-adaptive feature aggregation module, which synthesizes a pose-aware, viewpoint-invariant feature map to compensate for self-occlusions.

tions and the complex discrepancies between template and target geometries. Our core insight lies in leveraging the consistent representations of 2D foundation models pre-trained on large-scale datasets, modeling these foundation features via geometry-guided mechanisms. By explicitly aligning robust 2D features with 3D geometry, we aim to maintain a stable and informative conditioning signal, thereby facilitating generalizable deformation learning across diverse object categories.

**Geometry-Guided Feature Propagation.** Bridging 2D image semantics with 3D geometry introduces the challenge of partial observability. Image-based foundation features are inherently limited to the visible surface of the template, whereas deformation requires holistic guidance for the entire shape. To address this, we propose a geometry-guided propagation mechanism that diffuses semantic cues from visible regions to the full 3D structure. This approach relies on the insight that geometrically correlated regions often share semantic attributes, allowing us to infer features for occluded areas based on their structural affinity with visible counterparts. Specifically, we first extract features $\mathbf{F}_{\text{vis}} \in \mathbb{R}^{M \times D}$ corresponding to the subset of $M$ visible points on the template. Concurrently, a 3D encoder processes the complete point cloud to yield geometric embeddings $\mathbf{G} \in \mathbb{R}^{N \times d}$, capturing the structural context of all $N$ template points. These embeddings facilitate the computation of a pairwise affinity matrix, where the geometric similarity between a point $j$ in the full shape and a visible point $i$ is defined as:

$$S_{ji} = \frac{\mathbf{g}_j \cdot \mathbf{g}_i}{\|\mathbf{g}_j\| \|\mathbf{g}_i\|}, \qquad (2)$$

where $\mathbf{g} \in \mathbf{G}$ represents the geometric feature vector. Leveraging these affinities as a structural bridge, we propagate the

available 2D semantics to the entire shape via a weighted aggregation:

$$\mathbf{f}_j^{\text{complete}} = \sum_{i=1}^{M} \frac{\exp(S_{ji}/\tau)}{\sum_{k=1}^{M} \exp(S_{jk}/\tau)} \mathbf{f}_i^{\text{vis}}. \qquad (3)$$

This formulation enables occluded points to acquire semantic representations from geometrically related visible points, yielding the comprehensive feature set $\mathbf{F}_{\text{complete}} \in \mathbb{R}^{N \times D}$. This provides the deformation network with spatially consistent conditioning signals regardless of initial visibility.

**Semantic-Aware Feature Alignment.** While the propagated template features $\mathbf{F}_{\text{complete}}$ provide comprehensive semantic guidance regarding the template structure, effective deformation necessitates establishing robust correspondences between the target object and the template. The target image features $\mathbf{F}_{\mathcal{T}}$ encode the desired shape semantics within a 2D patch-based representation, whereas the template features operate on 3D point embeddings. To bridge this gap, we introduce an alignment module that leverages cross-attention to dynamically retrieve target semantics relevant to each template point. Specifically, we formulate the template features $\mathbf{F}_{\text{complete}} \in \mathbb{R}^{N \times D}$ as queries and the flattened target image features $\mathbf{F}_{\mathcal{T}} \in \mathbb{R}^{HW \times D}$ as keys and values. This allows us to synthesize the aligned features by aggregating target information via:

$$\mathbf{F}_{\text{aligned}} = \text{softmax}\left(\frac{(\mathbf{F}_{\text{complete}}\mathbf{W}_Q)(\mathbf{F}_{\mathcal{T}}\mathbf{W}_K)^T}{\sqrt{D}}\right)(\mathbf{F}_{\mathcal{T}}\mathbf{W}_V), \qquad (4)$$

where $\mathbf{W}_Q, \mathbf{W}_K, \mathbf{W}_V$ are learnable projections. This mechanism enables template points to attend specifically to semantically corresponding regions in the target image,

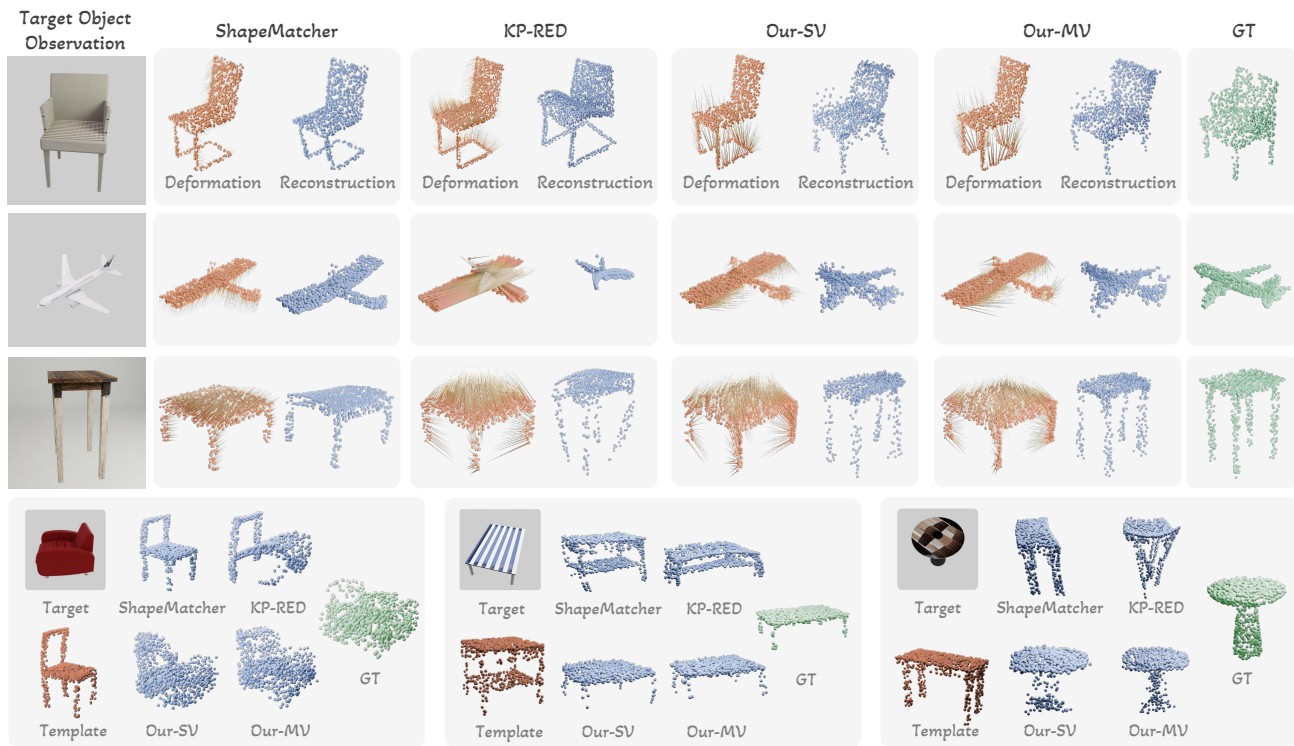

*Figure 3.* Qualitative comparison with existing shape deformation methods on novel target objects under the *Random Template* setting.

effectively overcoming geometric misalignments. The resulting aligned features $\mathbf{F}_{\text{aligned}}$ are further refined through self-attention layers to enforce local consistency, ultimately producing a conditioning signal $\mathbf{c}$ that guides the deformation network to predict shape changes that are both geometrically plausible and semantically faithful to the target.

## 3.2. View-Adaptive Feature Aggregation

While geometry-guided modeling effectively densifies the semantic signals into $\mathbf{F}_{\text{complete}}$, the underlying information remains rooted in a specific 2D observation. Consequently, these foundation features inherently encode viewpoint-dependent biases, where the same 3D structure may manifest differently in the feature space when observed from distinct camera poses. To address this, we propose a view-adaptive feature aggregation mechanism that dynamically incorporates camera pose embeddings to explicitly discriminate perspective-induced geometric misalignments from actual shape deformations, thereby disentangling viewpoint effects from intrinsic object semantics. This enables a viewpoint-invariant representation, providing stable guidance for the deformation learning.

### 3.2.1. MULTI-VIEW CAMERA POSE ENCODING

To align the multi-view features within a unified geometric context, we establish a canonical reference frame cen-

tered on the template view most semantically consistent with the target. Formally, given a set of template views $\mathcal{V}_{\mathcal{S}} = \{I_{\mathcal{S}}^k\}_{k=1}^K$ and their camera extrinsics $\{\mathbf{E}_{\mathcal{S}}^k\}_{k=1}^K$, we identify the primary view $I_{\mathcal{S}}^*$ by maximizing the foundation feature cosine similarity with the target image $I_{\mathcal{T}}$. The remaining views $\{I_{\mathcal{S}}^j\}_{j \neq *}$ serve as auxiliary sources, offering complementary structural details from varying viewpoints.

To capture the spatial configuration of the auxiliary views relative to the primary anchor, we compute the relative camera transformations. Specifically, we define the relative pose $\mathbf{P}_{\text{rel}}^k$ of the $k$-th view with respect to the primary view $*$ as:

$$\mathbf{P}_{\text{rel}}^k = (\mathbf{E}_{\mathcal{S}}^*)^{-1}\mathbf{E}_{\mathcal{S}}^k, \tag{5}$$

where $\mathbf{E}_{\mathcal{S}}$ represents the camera-to-world transformation matrix. This formulation explicitly encodes the geometric offset between the auxiliary and primary perspectives. We flatten the rotation and translation components of $\mathbf{P}_{\text{rel}}^k$ into a vector $\mathbf{p}^k \in \mathbb{R}^{12}$ and project it into the feature space:

$$\mathbf{e}^k = \mathbf{W}_{\text{pose}}\mathbf{p}^k + \mathbf{b}_{\text{pose}}, \tag{6}$$

where $\mathbf{W}_{\text{pose}} \in \mathbb{R}^{D \times 12}$ and $\mathbf{b}_{\text{pose}} \in \mathbb{R}^D$ are learnable parameters. The resulting embeddings $\mathbf{e}^k$ provide the network with viewpoint-invariant geometric cues, facilitating the disentanglement of pose-induced discrepancies from intrinsic shape deformations.

*Table 1.* Performance on seen and unseen categories with retrieved vs. random templates. Our-SV and Our-MV refer to our method using single-view template and multi-view template feature fusion, respectively. The best results are in **bold**.

| Methods | Seen Categories Unseen Objects | | | | | | Unseen Categories | | | | | |
| | Retrieved Template | | | Random Template | | | Retrieved Template | | | Random Template | | |
| | CD $(10^{-3})\downarrow$ | EMD $(10^{-2})\downarrow$ | S-IoU $(\%)\uparrow$ | CD $(10^{-3})\downarrow$ | EMD $(10^{-2})\downarrow$ | S-IoU $(\%)\uparrow$ | CD $(10^{-3})\downarrow$ | EMD $(10^{-2})\downarrow$ | S-IoU $(\%)\uparrow$ | CD $(10^{-3})\downarrow$ | EMD $(10^{-2})\downarrow$ | S-IoU $(\%)\uparrow$ |
|---|---|---|---|---|---|---|---|---|---|---|---|---|
| ShapeMatcher (Di et al., 2024) | 5.92 | 6.43 | 40.47 | 13.02 | 8.82 | 34.36 | - | - | - | - | - | - |
| KP-RED (Zhang et al., 2024) | 3.05 | 5.23 | 46.73 | 5.10 | 6.35 | 42.05 | - | - | - | - | - | - |
| Our-SV | 2.45 | 4.76 | 48.45 | 2.61 | 4.94 | 46.78 | 4.31 | 5.84 | 50.31 | 4.94 | 6.15 | 49.36 |
| Our-MV | **2.38** | **4.69** | **48.79** | **2.46** | **4.86** | **47.31** | **3.69** | **5.42** | **52.38** | **4.24** | **5.65** | **52.57** |

### 3.2.2. Pose-Aware Cross-View Feature Aggregation

Building upon the pose-encoded representations, we introduce an aggregation strategy designed to distill viewpoint-invariant semantics while exploiting complementary structural details from diverse orientations.

To explicitly inject geometric context, we first modulate the semantic features with their corresponding pose embeddings. For each visible point $i$ in the primary view, the feature $\mathbf{f}_{\text{partial}}^i \in \mathbb{R}^D$ is added to the canonical pose embedding:

$$\tilde{\mathbf{f}}_{\text{partial}}^i = \mathbf{f}_{\text{partial}}^i + \mathbf{e}^*, \tag{7}$$

where $\mathbf{e}^*$ represents the embedding of the canonical reference frame. Similarly, for the $M$ visible points in each auxiliary view $j$, we modulate their features $\mathbf{f}_{\text{aux}}^{j,i}$ with the relative pose embedding $\mathbf{e}^j$:

$$\tilde{\mathbf{f}}_{\text{aux}}^{j,i} = \mathbf{f}_{\text{aux}}^{j,i} + \mathbf{e}^j. \tag{8}$$

This operation aligns the semantic features with their geometric acquisition conditions.

To further enrich the primary view with context from auxiliary perspectives, we employ a cross-view attention mechanism. We designate the pose-modulated primary features as queries, while the concatenation of all view features serves as the keys and values. This configuration allows the primary view to selectively query semantically relevant regions across all available viewpoints. Formally, we stack all modulated features into a unified memory bank $\mathbf{F}_{\text{all}} \in \mathbb{R}^{(K \times M) \times D}$ and compute:

$$\mathbf{F}_{\text{fused}} = \text{Attention}(\mathbf{Q} = \tilde{\mathbf{F}}_{\text{primary}}, \mathbf{K} = \mathbf{F}_{\text{all}}, \mathbf{V} = \mathbf{F}_{\text{all}}), \tag{9}$$

where $\tilde{\mathbf{F}}_{\text{primary}} \in \mathbb{R}^{M \times D}$ denotes the matrix of primary features. The final aggregated representation is obtained via a residual connection:

$$\tilde{\mathbf{F}}_{\text{partial}} = \mathbf{F}_{\text{fused}} + \tilde{\mathbf{F}}_{\text{primary}}. \tag{10}$$

This formulation preserves the semantic integrity of the primary view while integrating complementary cues from auxiliary angles. The resulting features $\tilde{\mathbf{F}}_{\text{partial}}$ are robust to viewpoint variations and serve as the input for the subsequent geometry-guided propagation module.

### 3.3. Training Objectives

Training the deformation network requires supervising the predicted flow field to ensure both geometric accuracy and structural preservation. The primary objective is the flow matching loss $\mathcal{L}_{\text{FM}}$, which aligns the predicted deformation field $v_\theta(\mathbf{x}_t, t, \mathbf{c})$ with the target velocity field derived from the interpolation path. To prevent artifacts such as non-smooth deformations or excessive distortions, we incorporate several geometric regularizers: Chamfer distance loss $\mathcal{L}_{\text{CD}}$ for global shape alignment, Laplacian smoothness loss $\mathcal{L}_{\text{Lap}}$ for local continuity, ARAP loss $\mathcal{L}_{\text{ARAP}}$ for preserving local rigidity (Sorkine & Alexa, 2007), a regularization term $\mathcal{L}_{\text{reg}}$ to constrain deformation magnitude, and a silhouette loss $\mathcal{L}_{\text{sil}}$ for multi-view geometric supervision. The final training objective combines these terms:

$$\begin{aligned} \mathcal{L} = &\lambda_{\text{FM}}\mathcal{L}_{\text{FM}} + \lambda_{\text{CD}}\mathcal{L}_{\text{CD}} + \lambda_{\text{Lap}}\mathcal{L}_{\text{Lap}} \\ &+ \lambda_{\text{ARAP}}\mathcal{L}_{\text{ARAP}} + \lambda_{\text{reg}}\mathcal{L}_{\text{reg}} + \lambda_{\text{sil}}\mathcal{L}_{\text{sil}}, \end{aligned} \tag{11}$$

where $\lambda.$ are weights balancing the contribution of each term. Please refer to Sec. A.1 for more details on training objectives.

## 4. Experiments

In this section, we conduct extensive experiments to evaluate the effectiveness of our proposed method, including comparisons with deformation-based and 3D generative methods, ablation studies, and demonstrations of downstream applications.

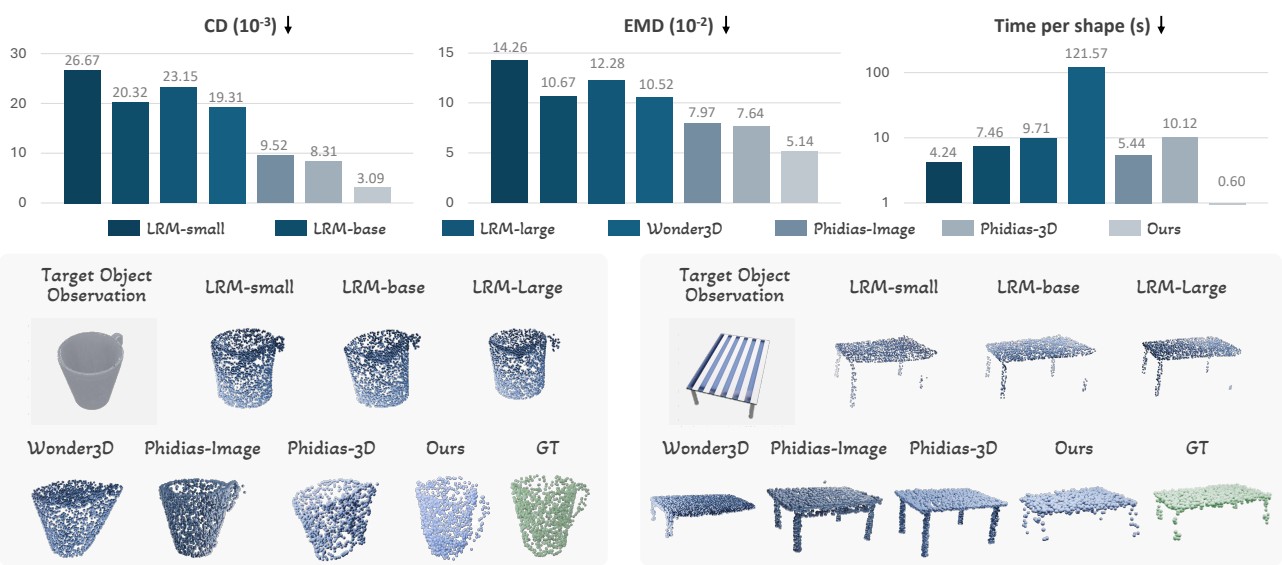

*Figure 4.* Quantitative and qualitative comparisons with existing 3D generative methods on single-view shape reconstruction. LRM-small, LRM-base, and LRM-large denote different model sizes. Phidias-Image and Phidias-3D refer to models conditioned solely on target images and those conditioned on additional 3D templates, respectively.

## 4.1. Experimental Setting

**Datasets.** Following prior works (Uy et al., 2021; Di et al., 2024; Zhang et al., 2024), we use ShapeNetv2 (Chang et al., 2015) as the training source and consider seven seen categories: chair, table, airplane, car, cabinet, bowl and bottle. For each category, we randomly sample 500 shape models, from which 50 models are further randomly selected as template objects; the remaining models are randomly split into training and test targets with a 9:1 ratio. To evaluate generalization to unseen categories, we additionally use OakInk (Yang et al., 2022) and select four categories: mug, teapot, lotion pump, and camera. For each category, we randomly choose 5 objects as templates and 20 objects as test targets, covering both real and synthetic instances. For data generation, we render RGB observations by sampling camera viewpoints on an object-centered upper hemisphere. We uniformly sample 16 camera poses around the template object to obtain multi-view template images, while for each target object, we randomly sample one viewpoint to obtain a single target observation, which naturally introduces challenges such as self-occlusion or partial visibility.

**Competing Methods and Evaluation Metrics.** We compare our approach against two representative deformation learning methods, ShapeMatcher (Di et al., 2024) and KP-RED (Zhang et al., 2024), and three 3D generative methods, LRM (Hong et al., 2024), Wonder3D (Long et al., 2024), and Phidias (Wang et al., 2025). We quantitatively evaluate the performance using Chamfer Distance (CD), Earth Mover's Distance (EMD), and silhouette IoU (S-IoU). Please refer to Sec. A.2 and Sec. A.3 for more details.

## 4.2. Main Results

We assess different methods on novel objects to evaluate shape deformation learning quality, employing two template selection strategies for each target observation. The *Retrieved Template* setting selects the template from the library that maximizes the cosine similarity in DINOv3 features between the canonical view and the target image. The *Random Template* setting selects a template arbitrarily and may introduce large shape variations. Regarding the baselines ShapeMatcher and KP-RED, we train separate models for each category while setting the target occlusion ratio to 50% and utilizing ground truth depth values to recover the target partial point cloud. In contrast, our method employs a unified model across all categories without relying on target observation depth information. We ensure fairness by using identical training data and testing pairs across all methods.

**Evaluation on Large Shape Variations.** Table 1 presents quantitative results on novel objects within seen and unseen categories. As can be seen, our method consistently achieves lower CD and EMD scores alongside higher S-IoU values to demonstrate superior deformation quality. Notably, the performance of the two baselines drops significantly when using random templates, whereas our method maintains performance levels similar to the retrieved template setting. This resilience shows that our geometry-guided modeling effectively captures the intricate similarity between the template and target observation to enhance robustness against shape variations. Figure 3 illustrates qualitative comparisons under the *Random Template* setting. It can be observed that baseline methods struggle to match the target when sig-

nificant topological changes occur, such as transforming a four-legged chair into a sofa or simplifying a two-tier table into a single-layer one. In contrast, our method handles these structural variations effectively.

**Evaluation on Unseen Categories.** For the challenging scenario where both template and target belong to unseen categories, ShapeMatcher and KP-RED are not applicable to this setting due to their requirement for per-category training. Consequently, we only evaluate our method and the results indicate that it continues to achieve high-quality deformations and demonstrates strong generalization capabilities for unseen categories. Figure 5 presents qualitative results of our method on unseen categories. Please refer to Sec. A.4 and Sec. A.6 for more results.

**Evaluation on Viewpoint Variations.** During dataset construction, we randomly sampled viewpoints around the target shape to render observations, inevitably introducing challenging scenarios such as severe self-occlusion. Figure 4 presents comparisons with 3D generative methods. As observed, when distinctive structures like a mug handle or table legs are partially occluded in the input view, the baselines tend to generate implausible shapes, often failing to reconstruct these missing parts. In contrast, our proposed deformation learning approach effectively leverages geometric priors from the template, preserving the essential topology of the template and ensuring the generated shape remains semantically plausible and consistent with the category.

### 4.3. Ablation Study

In this section, we perform an ablation study to verify the necessity of each proposed module. All model variants were retrained on the same dataset and evaluated on novel objects to assess deformation learning performance. The experimental results are summarized in Table 2.

**(i)** Replacing the flow matching framework with a direct regression objective (w/o FM) resulted in a performance drop. This indicates that modeling the deformation as a continuous trajectory via flow matching is superior to deterministic regression for capturing complex shape variations. **(ii)** We demonstrated that our geometry-guided foundation feature modeling strategy is essential for effective deformation learning. We validated this through two specific experiments: (1) by assigning features only to visible projected points on the template and setting the remaining points to the mean value (w/o Prop.); (2) by removing the relational modeling entirely, where target features were globally pooled and broadcast via FiLM to condition the template features (w/o Rel.). In both scenarios, the performance deteriorated, indicating that our geometry-guided design effectively captures the intricate spatial and semantic relationships between the template and target foundation features for precise deformation. **(iii)** View-adaptive feature aggregation module

*Table 2.* Quantitative ablation study results on novel objects.

| Methods | Retrieved Template | | | Random Template | | |
|---|---|---|---|---|---|---|
| | CD | EMD | S-IoU | CD | EMD | S-IoU |
| | $(10^{-3})\downarrow$ | $(10^{-2})\downarrow$ | $(\%)\uparrow$ | $(10^{-3})\downarrow$ | $(10^{-2})\downarrow$ | $(\%)\uparrow$ |
| w/o FM | 2.66 | 4.87 | 45.78 | 2.74 | 4.95 | 43.57 |
| w/o Prop. | 2.74 | 4.96 | 44.19 | 2.95 | 5.18 | 41.10 |
| w/o Rel. | 2.56 | 4.85 | 45.36 | 2.70 | 5.06 | 44.67 |
| w/o PrimSel. | 2.64 | 4.87 | 44.50 | 2.84 | 5.08 | 44.40 |
| w/o PoseAware. | 2.60 | 4.87 | 44.98 | 2.79 | 5.06 | 44.47 |
| Our-MV | **2.38** | **4.69** | **48.79** | **2.46** | **4.86** | **47.31** |

further enhances deformation learning. We compared our full model against using a single-view template reference (our-SV), removing the primary view selection by using the mean of all features as the aggregation query (w/o PrimSel.), and excluding camera pose encoding by directly averaging multi-view features (w/o PoseAware.). All these variations led to suboptimal results. Notably, we observed that ineffective utilization of multi-view foundation features (w/o PrimSel. and w/o PoseAware.) can yield results inferior to using a single view (our-SV). This confirms that the proposed multi-view fusion module plays a crucial role in constructing a robust, holistic template representation that mitigates occlusion and viewpoint bias.

### 4.4. Applications

Robotic dexterous grasp generation serves as a fundamental prerequisite for complex manipulation in humanoid robotics (Shu et al., 2024; Duan et al., 2022). While existing approaches include analytical (Liu et al., 2021; Wang et al., 2023; Ma et al., 2025a) and generative (Liu et al., 2023c; Lu et al., 2024) techniques, transfer-based methods (Yang et al., 2022; Ma et al., 2025b) typically exhibit stronger generalization by adapting grasp representations from known templates to novel targets via learned correspondences. However, the success of such methods critically depends on the quality of the underlying shape mapping.

Addressing this challenge, a distinct advantage of our deformation-based framework is the inherent preservation of dense point-wise correspondence, which naturally facilitates downstream tasks such as dexterous grasp transfer. As illustrated in Figure 5, we leverage the predicted deformation fields to warp contact maps from templates to novel targets, guiding the optimization of grasp configurations for arbitrary robotic hands. This transfer-based dexterous grasp generation process requires only about 0.67 s for deformation prediction and 15 s for contact-based grasp recovery. We validate this capability in Isaac Gym simulations using the Shadow Hand on the OakInk dataset (Yang et al., 2022). Compared to existing dexterous grasp generation methods,

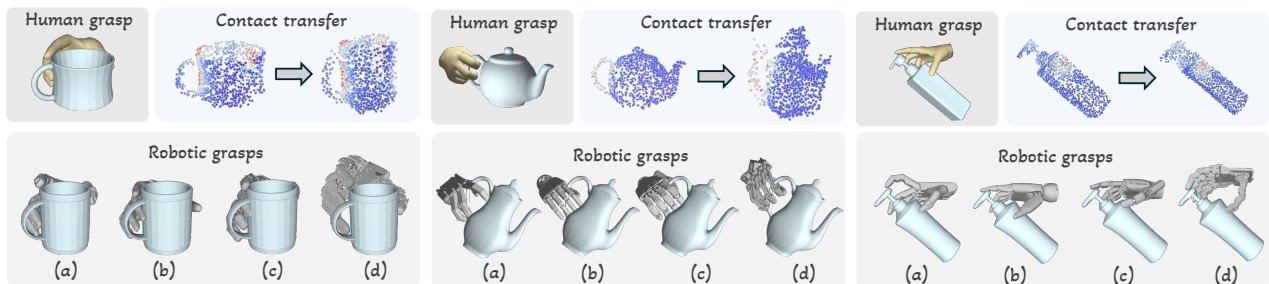

*Figure 5.* Qualitative results of contact map and grasp transfer for diverse objects across multiple robotic hands. Red and blue regions in the contact maps denote high and low contact values, respectively.

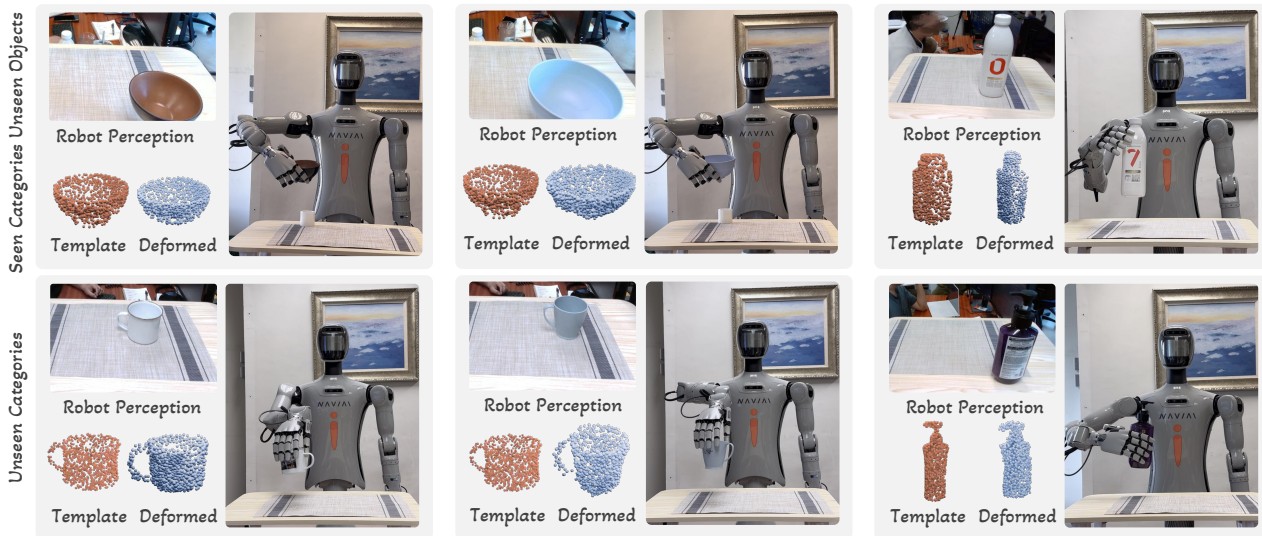

*Figure 6.* Qualitative results of generalizable dexterous manipulation in the real world.

our method achieves highly competitive grasp quality while significantly reducing computational overhead and bypassing the need for complete 3D target shapes. Furthermore, as shown in Figure 6, we demonstrate the practical applicability of our approach through real-world physical experiments on a NAVIAI AW-1 humanoid robot. Evaluated across four categories (*bowl*, *bottle*, *mug*, and *lotion pump*), our method achieves an impressive overall physical grasp success rate of 77%, demonstrating robust generalization to unseen objects in realistic settings. More detailed experimental setups, baseline comparisons, and quantitative metrics are provided in Sec. A.4.

## 5. Conclusion

In this paper, we presented a novel deformation learning framework for generalizable single-view 3D shape recovery. We introduced a geometry-guided feature modeling mechanism that enriches foundation features with template topology, enabling the model to capture fine-grained correspondences and handle complex shape variations across unseen categories. To mitigate performance degradation caused by viewpoint disparities, we developed a view-adaptive feature aggregation module that dynamically synthesizes a viewpoint-invariant template representation, ensuring consistent alignment between the fixed template and arbitrary target views. Extensive experiments demonstrate that our approach outperforms state-of-the-art methods in deformation and reconstruction quality, while also proving its practical efficacy in dexterous robotic manipulation tasks.

While our method shows promising results, single-view reconstruction remains an inherently ill-posed problem. When key regions of the target are fully occluded, the lack of explicit 2D deformation cues may lead to geometric discrepancies (please refer to Sec. A.8 for more details). In future work, we plan to extend our framework to support multi-view target inputs, thereby enriching the geometric information for more accurate deformation learning. Moreover, incorporating semantic priors from vision-language models can provide enhanced guidance for the reconstruction of ambiguous structures, which represents a promising direction for future investigation.

## Impact Statement

This paper presents a geometry-guided framework for monocular 3D object shape recovery via template deformation, leveraging semantically robust 2D foundation features to improve generalization across viewpoints and shape variations. The resulting capabilities may advance downstream research in areas such as affordance transfer and robotic manipulation, contributing to broader developments in autonomous systems. We do not foresee immediate severe risks or negative societal implications related to our contributions.

## Acknowledgment

This wok was supported in part by the National Natural Science Foundation of China Project No. 62322318, and in part by a grant from the NSFC/RGC Joint Research Scheme sponsored by the Research Grants Council of the Hong Kong Special Administrative Region, China and the National Natural Science Foundation of China (Project No. N_CUHK410/23), and in part by the Joint Funds of the National Natural Science Foundation of China (Grant No. U24A20128).

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

# A. Appendix

In this supplementary material, we provide additional details, experiments, and results to support the main paper:

## A.1. Details on Training Objectives

As formulated in the main paper, our total training objective $\mathcal{L}$ is a weighted combination of the flow matching loss and geometric regularization terms. In this section, we provide the detailed definitions for each component.

First, to supervise the continuous deformation trajectory, we minimize the difference between the predicted vector field $v_\theta$ and the target straight-line velocity field via the Flow Matching Loss ($\mathcal{L}_{\text{FM}}$). This loss is computed as an expectation over time $t$ and data pairs:

$$\mathcal{L}_{\text{FM}} = \mathbb{E}_{t,\mathbf{x}_0,\mathbf{x}_1} \left[ \|v_\theta(\mathbf{x}_t, t, \mathbf{c}) - (\mathbf{x}_1 - \mathbf{x}_0)\|_2^2 \right], \quad (12)$$

where $\mathbf{x}_t = (1-t)\mathbf{x}_0 + t\mathbf{x}_1$ represents the interpolated state. To ensure the deformed point cloud $\mathcal{P}_{\text{pred}}$ geometrically aligns with the target shape $\mathcal{P}_{\text{gt}}$, we employ the Chamfer Distance Loss ($\mathcal{L}_{\text{CD}}$), calculated as the symmetric sum of squared distances:

$$\mathcal{L}_{\text{CD}} = \sum_{\mathbf{p} \in \mathcal{P}_{\text{pred}}} \min_{\mathbf{q} \in \mathcal{P}_{\text{gt}}} \|\mathbf{p} - \mathbf{q}\|_2^2 + \sum_{\mathbf{q} \in \mathcal{P}_{\text{gt}}} \min_{\mathbf{p} \in \mathcal{P}_{\text{pred}}} \|\mathbf{q} - \mathbf{p}\|_2^2. \quad (13)$$

To prevent high-frequency artifacts and maintain local surface smoothness, we incorporate the Laplacian Smoothness Loss ($\mathcal{L}_{\text{Lap}}$). Let $\delta_i$ be the Laplacian coordinate of vertex $i$ (the difference between the vertex and the centroid of its neighbors). We minimize:

$$\mathcal{L}_{\text{Lap}} = \sum_i \|\delta_i^{(\text{pred})} - \delta_i^{(\text{src})}\|_2^2, \quad (14)$$

which encourages the local geometry of the deformed mesh to remain consistent with the source mesh topology. Furthermore, to preserve structural rigidity and avoid unnatural

*Table 3.* Quantitative comparison with different dexterous grasp generation methods. The best results are in **bold**.

| Methods | SR (%) ↑ | Pen. (mm) ↓ | Cov. (%) ↑ | Time (s) ↓ |
|---|---|---|---|---|
| **Analytical Method** | | | | |
| DFC (Liu et al., 2021) | 65.00% | 8.01 | 30.07% | >1000 |
| **Generative Methods** | | | | |
| ConGen (Liu et al., 2023c) | 47.59% | 3.05 | 23.19% | 17.1 |
| UGG (Lu et al., 2024) | 58.00% | 8.37 | **36.65%** | 76.0 |
| **Transfer-Based Methods** | | | | |
| Tink (Yang et al., 2022) | 61.96% | 4.60 | 27.91% | 87.6 |
| cmtDiff (Ma et al., 2025b) | 69.59% | **2.70** | 31.17% | 62.2 |
| Our-MV | **76.92%** | 2.86 | 28.39% | **15.7** |

distortions (e.g., shearing), we enforce the As-Rigid-As-Possible (ARAP) Loss ($\mathcal{L}_{\text{ARAP}}$) (Sorkine & Alexa, 2007). This term minimizes the deviation of local transformations from rigid rotations:

$$\mathcal{L}_{\text{ARAP}} = \sum_i \sum_{j \in \mathcal{N}(i)} w_{ij} \|(\mathbf{p}_i' - \mathbf{p}_j') - \mathbf{R}_i(\mathbf{p}_i - \mathbf{p}_j)\|_2^2, \quad (15)$$

where $\mathbf{p}, \mathbf{p}'$ denote vertex positions before and after deformation, $\mathcal{N}(i)$ are neighbors of vertex $i$, $w_{ij}$ are cotangent weights, and $\mathbf{R}_i$ is the optimal rotation matrix for the local neighborhood.

We also apply a Regularization Loss ($\mathcal{L}_{\text{reg}}$) to constrain the magnitude of the deformation vectors $\mathbf{d}_i$ and encourage minimal necessary displacement:

$$\mathcal{L}_{\text{reg}} = \frac{1}{N} \sum_{i=1}^{N} \|\mathbf{d}_i\|_2^2. \quad (16)$$

Finally, to leverage multi-view supervision, we utilize the Silhouette Loss ($\mathcal{L}_{\text{sil}}$) by rendering the deformed shape into 2D silhouettes $S_{\text{pred}}^{(k)}$ from $K$ viewpoints and comparing them with ground truth masks $S_{\text{gt}}^{(k)}$:

$$\mathcal{L}_{\text{sil}} = \sum_{k=1}^{K} \|S_{\text{pred}}^{(k)} - S_{\text{gt}}^{(k)}\|_2^2. \quad (17)$$

## A.2. Details on Evaluation Metrics

We quantitatively evaluate the reconstruction quality using three standard metrics: Chamfer Distance (CD), Earth Mover's Distance (EMD), and Silhouette IoU (S-IoU). **Chamfer Distance (CD)** measures the geometric accuracy between the predicted point cloud $\mathcal{P}$ and the ground truth $\mathcal{G}$ without requiring point-to-point correspondence. We compute the symmetric Chamfer distance using the L2 norm, defined as the sum of the average nearest neighbor distances in both directions: $\text{CD}(\mathcal{P}, \mathcal{G}) = \frac{1}{|\mathcal{P}|} \sum_{\mathbf{p} \in \mathcal{P}} \min_{\mathbf{g} \in \mathcal{G}} \|\mathbf{p} -$

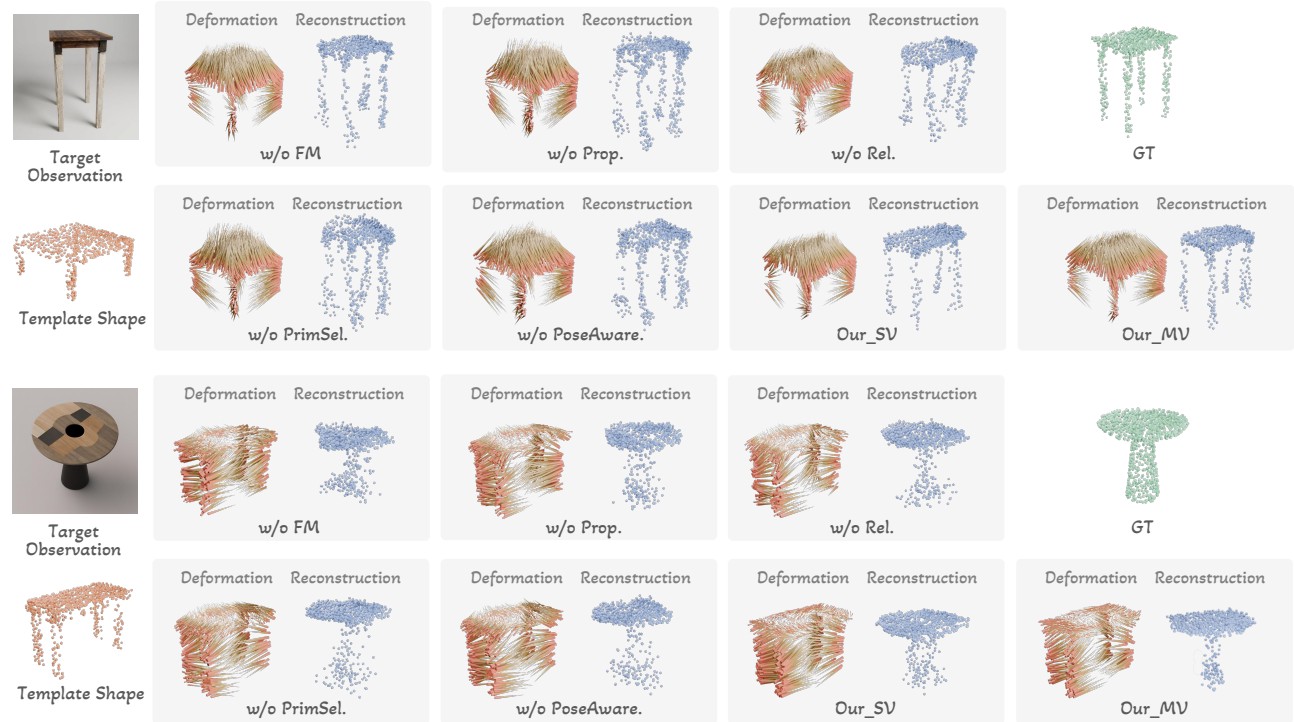

*Figure 7.* Qualitative comparison of deformation and reconstruction results under different ablation settings.

$\mathbf{g}\|_2^2 + \frac{1}{|\mathcal{G}|}\sum_{\mathbf{g}\in\mathcal{G}}\min_{\mathbf{p}\in\mathcal{P}}\|\mathbf{g}-\mathbf{p}\|_2^2$. Lower CD values indicate better surface alignment. **Earth Mover's Distance (EMD)** captures both geometry and point density by solving an optimal transport problem. It seeks a bijection $\phi:\mathcal{P}\to\mathcal{G}$ that minimizes the average distance: $\text{EMD}(\mathcal{P},\mathcal{G}) = \min_\phi \frac{1}{|\mathcal{P}|}\sum_{\mathbf{p}\in\mathcal{P}}\|\mathbf{p}-\phi(\mathbf{p})\|_2$, where lower values imply a more accurate reconstruction of the shape distribution. Finally, **Silhouette IoU (S-IoU)** evaluates the visual fidelity from a specific 2D perspective. We render binary masks $M$ from the observed viewpoint and calculate the Intersection-over-Union between the predicted and ground truth projections: S-IoU $= \frac{|M_{\text{pred}}\cap M_{\text{gt}}|}{|M_{\text{pred}}\cup M_{\text{gt}}|}$. Higher S-IoU values indicate better structural correctness in the projected 2D space.

### A.3. Implementation Details

We represent each template shape as a point cloud with $N = 1024$ points and render $K = 16$ support views, with images at resolution $H = W = 512$; for partial observations, we set the number of visible points to $M = 512$. We extract patch-level features from the final transformer block (layer 12) of a pretrained DINOv3 ViT-B/16 model, yielding a feature dimension of $D = 768$. During geometry-guided feature lifting, we adopt Point-MAE (Pang et al., 2023) to capture structural relationships within the complete point cloud, using a 12-layer Transformer encoder with embedding dimension 384 and 6 attention heads, and

a decoder that outputs point-wise features of dimension $d = 256$. For both the feature alignment module and the multi-view camera feature fusion module, we integrate contextual information into a 512-dimensional query stream via an initial cross-attention block, followed by a stack of self-attention layers with 8 heads for feature refinement. We adopt PointTransformerV3 (Wu et al., 2024) as our deformation backbone, which employs a hierarchical encoder-decoder architecture with serialized attention mechanisms to process point clouds organized into patches. We set the loss weights to $\lambda_{FM} = \lambda_{Lap} = \lambda_{ARAP} = \lambda_{reg} = 1.0$, $\lambda_{CD} = 100.0$, and $\lambda_{sil} = 5.0$. We train the model from scratch using Adam with batch size 8, 4 dataloader workers, and an initial learning rate of $10^{-5}$ annealed at 50% of the training epochs using a cosine schedule. All models are trained for 100 epochs on two NVIDIA H800 GPUs, which takes approximately 36 hours.

### A.4. Additional Application Results

In this section, we validate the practical utility of our framework by leveraging the predicted template-to-target deformation fields for dexterous grasp transfer. Robotic dexterous grasp generation aims to synthesize stable and physically plausible hand poses for manipulating objects, serving as a fundamental prerequisite for enabling complex dexterous manipulation in humanoid robotics (Shu et al., 2024; Duan et al., 2022). Existing approaches can be broadly catego-

*Table 4.* Quantitative results under cross-category template assignments. We evaluate our method under three template-selection regimes: (i) Same sub-category, where the template is randomly drawn from a same sub-category (ii) Same super-category, where the template is drawn from a different sub-category within the same super-category, and (iii) Cross super-category, where the template is sampled from a different super-category.

| Test category | Ours | | | | | | Phidias-3D | |
|---|---|---|---|---|---|---|---|---|
| | Same-sub | | Same-super | | Cross-super | | Same-sub | |
| | CD $(10^{-3})\downarrow$ | EMD $(10^{-2})\downarrow$ | CD $(10^{-3})\downarrow$ | EMD $(10^{-2})\downarrow$ | CD $(10^{-3})\downarrow$ | EMD $(10^{-2})\downarrow$ | CD $(10^{-3})\downarrow$ | EMD $(10^{-2})\downarrow$ |
| Furniture | 3.41 | 5.66 | 7.88 | 7.21 | 15.86 | 10.31 | 11.22 | 8.34 |
| Container | 2.21 | 4.50 | 7.51 | 7.52 | 8.81 | 8.39 | 8.07 | 7.45 |
| Vehicle | 1.36 | 3.88 | 5.94 | 7.13 | 11.57 | 9.90 | 5.78 | 6.93 |
| Overall | 2.56 | 4.88 | 7.36 | 7.30 | 12.42 | 9.52 | 8.31 | 7.64 |

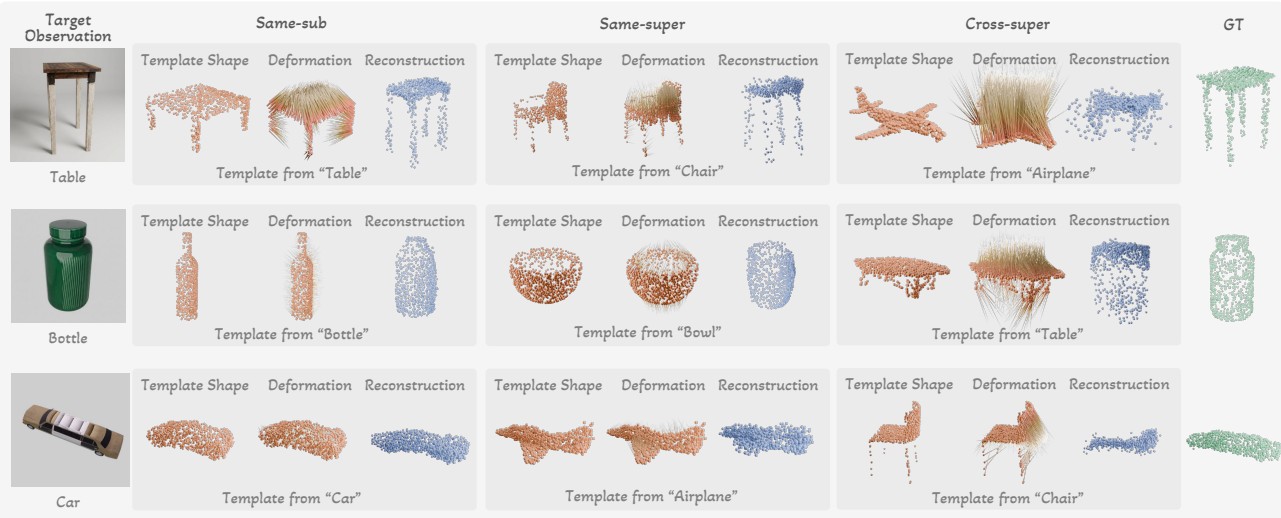

*Figure 8.* Qualitative results under cross-category template assignments. For each test instance, we show reconstructions produced by our method using Same sub-category templates (left), Same super-category templates (center), and Cross super-category templates (right). The selected template category is indicated in each panel. As the template becomes less related to the target object, reconstructions progressively deviate from the correct part structure and geometry, illustrating the model's failure mode under severe template mismatch.

rized into analytical (Liu et al., 2021; Wang et al., 2023), generative (Liu et al., 2023c; Lu et al., 2024), and transfer-based methods (Yang et al., 2022; Ma et al., 2025b). Among them, transfer-based methods typically exhibit stronger generalization capabilities by adapting grasp-related representations from known templates to novel targets via learned correspondences. However, the success of such methods critically depends on the quality of the underlying shape mapping. Here, we demonstrate that our framework provides an effective solution for transferring known human grasps defined on template shapes to diverse, unseen target objects, supporting arbitrary multi-finger robot hands.

### A.4.1. SIMULATION EXPERIMENTS

**Methodology.** Specifically, following the protocols established in (Liu et al., 2023c; Ma et al., 2025b), we represent hand-object interaction using an object-centric contact map. We utilize objects from the OakInk dataset (Yang et al., 2022), which provides parametric human grasps as templates, and compute a contact map on each template shape. This map is defined as a point-wise contact probability in $[0, 1]$, where a higher value indicates a higher likelihood of hand contact. Given an RGB observation of a target object, our model predicts a template-to-target deformation field to reconstruct the target shape. Leveraging the induced point-wise correspondence, we directly warp the template contact map onto the reconstructed target geometry. Finally, we

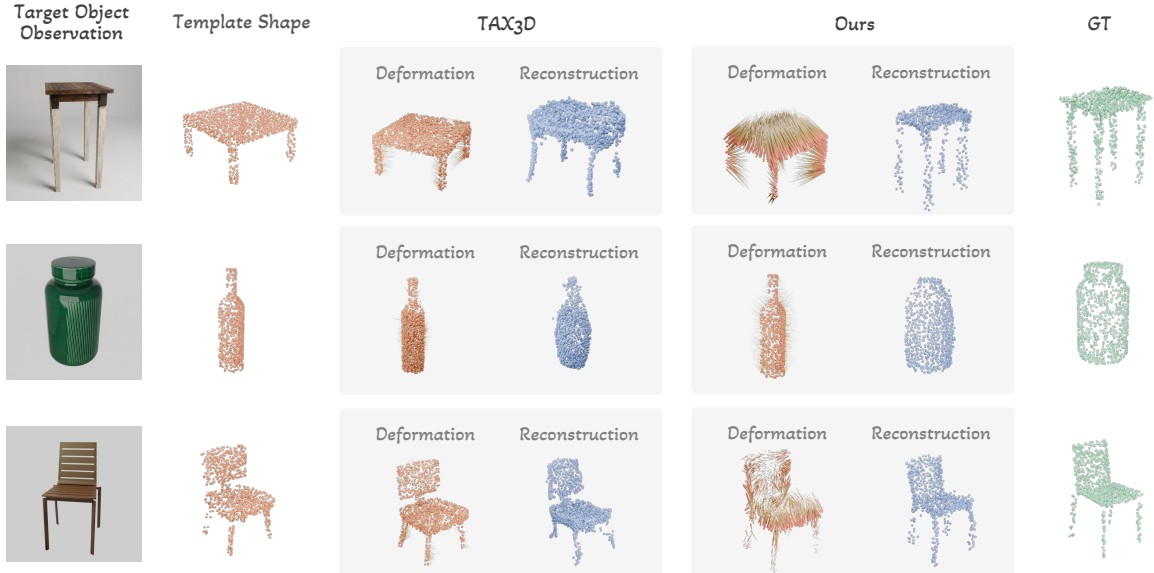

*Figure 9.* Qualitative comparison with TAX3D on shape deformation. Our method achieves superior geometric consistency and demonstrates greater robustness to large template-target variations compared to TAX3D.

recover the grasp configuration for various robotic hands on the target object by optimizing for the transferred contact information.

In detail, we reconstruct the robotic hand mesh from joint angles $\mathbf{q}$ using differentiable forward kinematics. Beyond the dense contact map, we also leverage sparse contact cues by identifying the five fingertip positions relative to the template shape. Using the predicted deformation field, we map these keypoints to their corresponding target locations, denoted as $\mathcal{P}_{\text{tgt}} \in \mathbb{R}^{5 \times 3}$. We then iteratively update $\mathbf{q}$ by minimizing a composite loss function:

$$\mathcal{L}_{\text{syn}} = \lambda_{\text{cont}} E_{\text{cont}} + \lambda_{\text{tip}} E_{\text{tip}} \qquad (18)$$

where $E_{\text{cont}} = \mathcal{L}_{\text{MSE}}(\hat{\mathbf{c}}, \overline{\mathbf{c}})$ represents the mean squared error between the transferred contact map $\hat{\mathbf{c}}$ and the current map $\overline{\mathbf{c}}$ derived from $\mathbf{q}$. The term $E_{\text{tip}} = \sum_{i=1}^{5} \|\mathbf{p}_i(\mathbf{q}) - \mathbf{p}_{\text{tgt},i}\|_2^2$ enforces the Euclidean distance between the current fingertip positions $\mathbf{p}_i(\mathbf{q})$ and their target locations $\mathbf{p}_{\text{tgt},i}$.

**Evaluation Metrics.** To comprehensively evaluate grasp quality, we benchmark our method against state-of-the-art analytical, generative, and transfer-based approaches (Wei et al., 2024; Liu et al., 2023c; Ma et al., 2025b). Experiments are conducted using the Shadow Hand (sha, 2005) across three object categories unseen during training: *mug*, *teapot*, and *lotion pump*. Each category contains 10 target objects with 5 different template grasps. We employ three standard metrics to quantify the performance. **Success Rate (SR)** measures the ability to stably lift an object in the Isaac-Gym simulator (Makoviychuk et al., 2021). Under standard gravity ($-9.8m/s^2$), a grasp is deemed successful if the

object is lifted above 10 cm and shifts less than 2 cm after 60 steps. We report the average success rate over 10 trials with randomized object poses. **Penetration Depth (Pen.)** evaluates collision artifacts by measuring the maximum distance any object point penetrates the hand mesh. **Contact Coverage (Cov.)** assesses the percentage of hand vertices located within a $\pm 2$mm threshold of the object surface.

**Results.** Table 3 reports the quantitative results for dexterous grasp generation. As expected, previous transfer-based methods generally achieve higher grasp success rates and better grasp quality compared to generative baselines. However, they require access to complete target shapes to transfer contacts and typically incur substantial runtime (at least 60 s), which limits practical deployment. In contrast, our method predicts the template-to-target deformation directly from a single target RGB observation by modeling 3D shape variation with 2D foundation features. This enables rapid deformation inference (approx. 0.67 s) followed by a 15 s optimization to recover the grasp configuration, thus offering improved generalization and usability in realistic settings. Figure 5 in the main paper presents qualitative examples of grasp transfer onto diverse objects using multiple robotic hands (Shadow Hand, Inspire Hand, NAVIAI Hand, and a self-developed dexterous hand), demonstrating that our method provides accurate dense correspondence to effectively enhance dexterous grasp generation.

### A.4.2. REAL-WORLD ROBOTIC EXPERIMENTS

We further conducted real-world experiments to evaluate the effectiveness of our method in practical scenarios. The

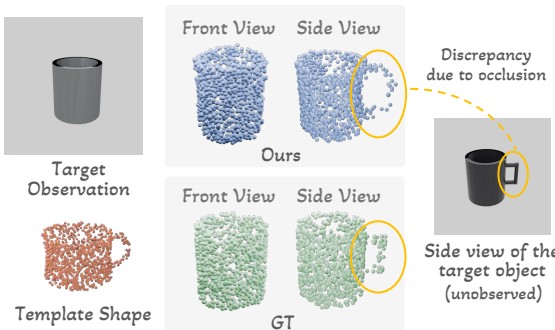

*Figure 10.* Failure Analysis. When key regions are fully occluded in the target observation, our method tends to preserve the template's corresponding structure. This leads to a discrepancy with the true target shape if the unobserved geometry differs from the template.

experiments were performed on a NAVIAI AW-1 humanoid robot equipped with a dexterous hand featuring 15 degrees of freedom (DoFs) and 6 active joints. As shown in Figure 6, the setup consists of a table placed in front of the robot.

Given the target observation captured by the robot's built-in camera, we first employ SAM3 (Carion et al., 2025) to segment the target object, which is then fed into our model to predict the deformation from the template. Subsequently, we transfer the human grasp from the template to the robotic hand following the optimization method described above. For grasp execution, we follow the protocol in (Wei et al., 2024): the robot arm first moves to the predicted 6-DoF pose of the hand's root, after which the dexterous hand actuates its joint angles based on the predicted grasp configuration.

We evaluated our method on four object categories: *bowl, bottle, mug, and lotion pump*. We conducted five trials for each object instance to compute the average success rate. A grasp is considered successful only if the robot can lift the object by at least 20 cm and maintain a stable hold for a duration of 10 seconds. Figure 6 illustrates qualitative results from the physical experiments, highlighting our method's ability to accurately deform template shapes to match real-world objects captured by robot perception. These results demonstrate effective generalization to novel objects and categories in real-world settings. Quantitative results show an overall success rate of 77% (60% for bowls, 90% for bottles, 80% for mugs, and 80% for lotion pumps), confirming the practical applicability of our approach and its robustness to real-world observations.

## A.5. Additional Comparison Results

In addition to the state-of-the-art deformation methods evaluated in the main paper, we further compare our approach with TAX3D (Cai et al., 2025), a recent diffusion-based deformation field generation method. Specifically, we set the template object as the initial action point cloud ($P_A$) and

*Table 5.* Quantitative comparison of shape deformation performance between our method and TAX3D on unseen objects.

| Method | CD ↓ | EMD ↓ | S-IoU ↑ | Time (s) ↓ |
|---|---|---|---|---|
| TAX3D (Cai et al., 2025) | 5.83 | 6.49 | 40.11 | 3.23 |
| Ours | **2.46** | **4.86** | **47.31** | **0.64** |

the target object as the anchor point cloud ($P_B$) to predict the per-point displacement flow from $P_A$ to $P_B$. We used the ground-truth depth to recover the target partial point cloud, consistent with the evaluation of other baselines in the main paper. We trained TAX3D from scratch on the same training dataset used for our method, following the default hyperparameter from its original paper.

Both quantitative and qualitative results are presented in Table 5 and Figure 9. As demonstrated, our method achieves lower CD and EMD errors, as well as better qualitative shape consistency after deformation. Because single-view target observations are inherently partial, using them directly as a condition (as TAX3D does) leads to information loss. In contrast, our method leverages the robust semantic similarity provided by 2D foundation models to guide 3D shape deformation learning. This makes our approach much more robust to large template-target shape variations and yields superior generalization to unseen objects. Additionally, our single-step rectified flow matching framework is computationally more efficient than the iterative denoising process required by the diffusion model in TAX3D.

## A.6. Additional Qualitative Results

Figure 11 presents additional deformation and reconstruction results on diverse objects, utilizing target observations either rendered from the test set or synthesized by the image generation model (Team et al., 2023). Despite significant shape variations between the template and target, our method yields smooth, consistent deformation fields and faithfully reconstructs the target geometry. Figure 12 illustrates the deformation and reconstruction outcomes on real-world unseen object categories. Additionally, we demonstrate the transfer of human grasps and template contact maps to the target via the estimated deformation fields, enabling the generation of robotic dexterous grasps across diverse robot hand models. Figure 7 presents the qualitative results under different ablation settings. It can be observed that removing any key component leads to varying degrees of performance degradation, such as discontinuous deformations, compromised structural integrity in the generated shapes, and poor reconstruction of unobserved regions. These visual artifacts confirm the necessity of each module for achieving high-quality deformation and reconstruction.

## A.7. Discussion on Robustness to Cross-Category Template Assignments

In the main paper, we demonstrated the effectiveness of our deformation-based approach at the category level. To further explore its generalization capabilities and systematically investigate the robustness of our framework against template mismatches, we evaluate its performance under cross-category template assignments. Specifically, we group the test objects into three broad super-categories: Furniture (*chair, table, cabinet*), Container (*bowl, bottle, mug, teapot, lotion pump*), and Vehicle (*car, airplane*). Without any retraining, we assess the model under three distinct template selection settings: 1) *Same-sub* (template from the exact same sub-category), 2) *Same-super* (template from a different sub-category but within the same super-category), and 3) *Cross-super* (template from a completely different super-category).

As illustrated in Table 4 and Figure 8, transitioning from the *Same-sub* to the *Same-super* setting leads to a moderate increase in CD and EMD. However, our method remains highly competitive with template-augmented 3D generative baselines, such as Phidias-3D. This demonstrates the model's robustness and highlights its underlying mechanism: while 2D observations provide strong target guidance, the template's topology acts as a structural regularizer during shape deformation. For instance, in *Same-super* cases (e.g., deforming a bowl template to a bottle target), the model successfully matches the overall silhouette of the bottle while preserving the open-top topology of the bowl. Conversely, moving to the *Cross-super* setting marks the failure threshold of our model. In these scenarios, the extreme topological discrepancy (e.g., deforming a car template to a cup) exceeds the inherent deformation capacity of the network, resulting in a substantial performance drop.

## A.8. Failure Analysis

Our method leverages the template shape, template 2D foundation features, and their correlation with the target's 2D features to enrich the shape representation. While this approach ensures robustness to viewpoint variations by preserving the template's structure in unobserved regions, it may fail to recover the accurate target geometry when the invisible parts differ significantly from the template. As shown in Figure 10, when the handle of the mug is totally occluded, our method successfully recovers the cup body but retains the handle structure of the template. This ambiguity is an inherent challenge in single-view reconstruction. In future work, we plan to extend our model to support multi-view target inputs, thereby achieving more accurate deformation learning and reconstruction.

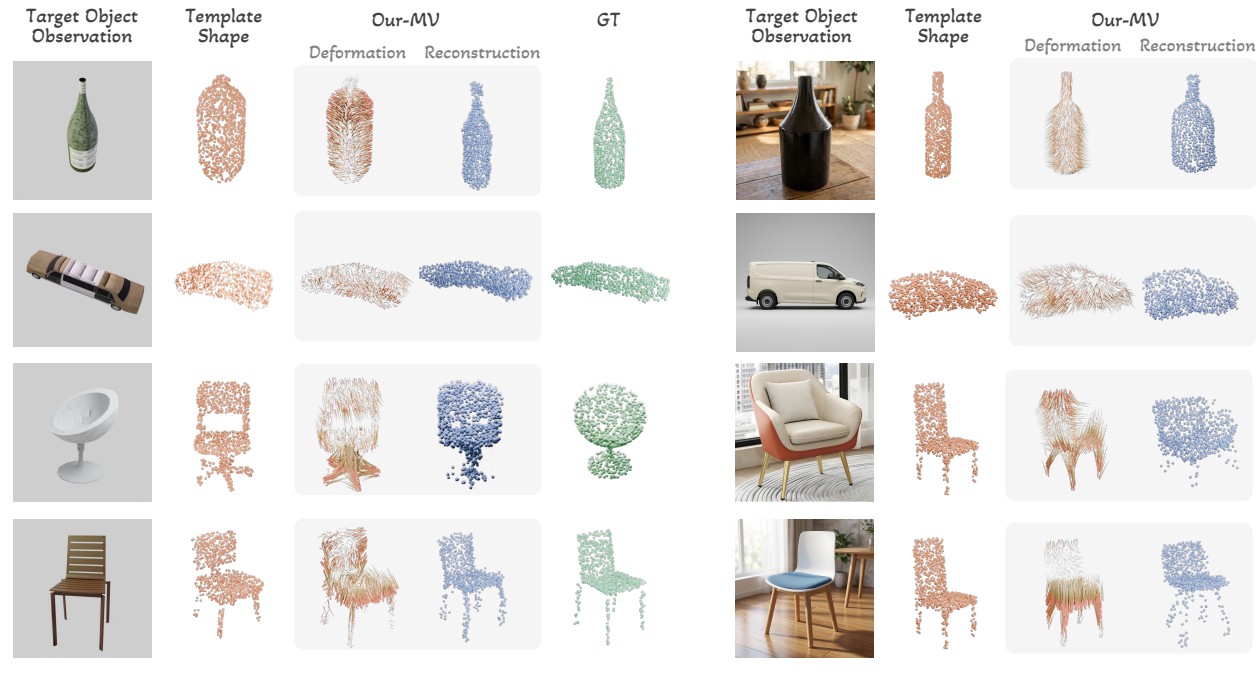

*Figure 11.* Visualization of shape and corresponding deformation field generated by our proposed method on diverse novel objects.

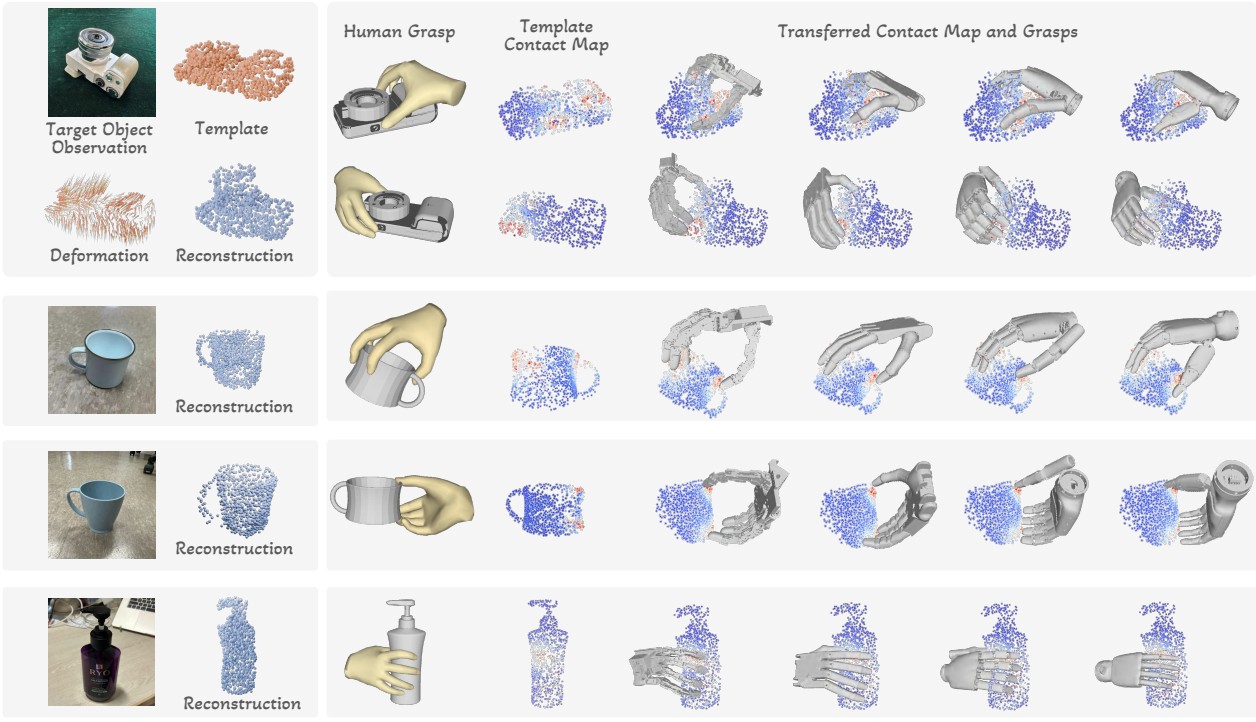

*Figure 12.* Qualitative results of deformation and reconstruction on real-world unseen object categories, along with transferred contact maps and robotic dexterous grasps.

