# OpenReview forum: "Geometry-Guided Modeling of Foundation Features Enables Generalizable Object Shape Deformation Learning"
_ICML.cc/2026/Conference — ICML 2026 regular_

### Official Review · Reviewer_3xKS · 2026-02-17

**Soundness:** 2
**Presentation:** 2
**Significance:** 2
**Originality:** 2
**Overall Recommendation:** 3
**Confidence:** 4

**Summary:**

The paper presents a deformation learning framework that reconstructs 3D objects by explicitly deforming a category-level shape template to match the target observation. The complex shape variations in deformation are addressed via a geometry-guided feature modeling mechanism that enriches foundation features with template topology to establish precise point-wise correspondences. The author also claims that the work is useful for downstream robotic manipulation tasks.

**Compliance With Llm Reviewing Policy:**

Affirmed.

**Key Questions For Authors:**

See weaknesses above.

**Limitations:**

No limitations were provided.

**Strengths And Weaknesses:**

Strengths:

- The paper studies a well-established problem in the field of 3D vision from a different perspective, by explicitly incorporating geometry into the flow-matching process when learning the deformation field.
- Incorporating the features into flow-matching using a cosine-similarity-based metric is an elegant formulation.
- The method seemed to outperform other baselines.

---
Weaknesses:

- The main problem with the paper is the lack of comparisons in the generation of the deformation field. Previous work such as [1] explored the use of DiT as the backbone, resulting in a flexible, multimodal per-point displacement field for deformable objects. The authors seemed to ignore other relevant works that achieve the same end result.
- The authors claimed the effectiveness of embodiment-transfer in robotic grasping tasks. However, the author only showed limited results. While this is an interesting application, a large body of previous work has dealt with grasping, which the authors again did not compare or mention.

[1] TAX3D: Non-Rigid Relative Placement through Dense Diffusion

---

> ### Author Rebuttal · Authors · 2026-03-31
>
> Dear Reviewer 3xKS, we appreciate your time and effort in reviewing our work. We are encouraged that you consider our approach an "elegant formulation" that tackles "a well-established problem" "from a different perspective," can "outperform other baselines," and is "useful for robotic manipulation tasks." Our detailed, point-by-point responses are provided below:
>
> ---
>
> **W1: Comparison with one additional deformation method TAX3D**
>
> **A1**: We sincerely thank the reviewer for the insightful suggestion to strengthen our experimental comparisons. In our main paper, we primarily compared our method against state-of-the-art deformation learning methods KP-RED and ShapeMatcher. Following your suggestion, we further adapted the recent diffusion-based deformation field generation method TAX3D to our shape deformation setting. Specifically, we set the template object as the initial action point cloud and the target object as the anchor point cloud to predict the per-point displacement flow. We used the ground-truth depth to recover the target partial point cloud, consistent with the evaluation of other baselines in our main paper. We trained TAX3D from scratch on the same training dataset used for our method, following the default hyperparameters from its original paper.
>
> As shown in the Table R3 and [Figure R5](https://anonymous.4open.science/r/anony-k5a5X4t/FigureR5.png), our method achieves lower CD and EMD errors, as well as better qualitative shape consistency after deformation. Because single-view target observations are inherently partial, using them directly as a condition leads to information loss. In contrast, our method leverages the robust semantic similarity provided by 2D foundation models to guide 3D shape deformation learning. This makes our approach much more robust to large template-target shape variations and yields superior generalization to unseen objects. We will include this experiment and discussion in the revised manuscript. Thank you again for helping us strengthen our evaluation.
>
> Table R3. Quantitative comparison of shape deformation performance between our method and TAX3D on unseen objects.
>
> | Method | CD ($10^{-3}$) $\downarrow$ | EMD ($10^{-2}$) $\downarrow$ | S-IoU (%) $\uparrow$ | Time per shape (s) |
> | ------ | --------------------------- | ---------------------------- | -------------------- | ------------------ |
> | Tax3D  | 5.83                        | 6.49                         | 40.11                | 3.23               |
> | Ours   | **2.46**                    | **4.86**                     | **47.31**            | **0.64**           |
>
> ---
>
> **W2: More discussion on embodiment-transfer in robotic grasping tasks**
>
> **A2**: We appreciate the reviewer’s interest in our dexterous grasping application and agree that grasping has a large and active literature. One of the key advantages of our deformation-based framework is the inherent preservation of point-wise correspondence, which naturally facilitates applications such as embodiment transfer for robotic dexterous grasp generation.
>
> We apologize if the presentation in the main text gave the impression of limited results. Due to page limits, we could only include Figures 1 and 5 in the main manuscript. However, we have provided extensive experimental details, comprehensive comparisons with prior work, and both quantitative and qualitative analyses in the **appendix (Sec. A.4, pages 14–16)**.
>
> Regarding comparisons with previous work, we explicitly position our approach within the realm of *transfer-based dexterous grasp generation*. In this setup, we follow previous works by representing the template grasp as an object-centric contact map and transferring it to diverse, novel target objects using our predicted deformation field. In **Table 3 (page 13)**, we comprehensively compare our approach against existing state-of-the-art analytical methods, generative methods, and other transfer-based methods in terms of grasping success rate, physical plausibility, and computational efficiency. According to these results, compared to existing transfer-based methods, our approach demonstrates improved usability, generalizability, and efficiency in realistic settings.
>
> Furthermore, **Figure 10 (page 17)** shows qualitative results of transferring contact maps and dexterous grasps to real-world unseen objects. We also deployed our method on a real-world humanoid robot equipped with a dexterous hand. The quantitative grasping success rates across four object categories are reported in **lines 836–837 (page 16)**, accompanied by qualitative real-world execution results in **Figure 6 (page 14)**.
>
> We will ensure that the references to these supplementary baselines and real-world experiments are made much more prominent in the revised main text. Thank you again for your interest and careful attention to this application.

---

> > ### Author Rebuttal · Reviewer_3xKS · 2026-04-03
> >
> > Thank you for your hard work. While the new results addressed my questions, he presentation of the paper could still be improved.

---

> > > ### Author Response · Authors · 2026-04-05
> > >
> > > Dear Reviewer 3xKS, thank you so much for your time and for acknowledging that our new results and responses have addressed your questions! We deeply appreciate your constructive feedback, which has been instrumental in helping us improve our work. To ensure we have thoroughly addressed your initial questions and to clarify how we will enhance the manuscript, we would like to briefly summarize our action plan based on your valuable suggestions.
> > >
> > > 1. **Highlighting the comparison with TAX3D**
> > >
> > > Following your insightful suggestion, in addition to the state-of-the-art deformation methods already evaluated in our original manuscript, we have compared our approach with the additional baseline TAX3D both qualitatively and quantitatively. As demonstrated in our rebuttal (Table R3 and [Figure R5](https://anonymous.4open.science/r/anony-k5a5X4t/FigureR5.png)), our method achieves superior accuracy (e.g., lower CD and EMD) and faster inference speeds. This further validates that leveraging the robust semantic similarity from 2D foundation models to guide 3D shape deformation significantly enhances robustness against large template-target variations and yields superior generalization to unseen target observations. In the revised manuscript, we will explicitly include this new baseline comparison and expand the discussion on deformation field generation in the main text.
> > >
> > > 2. **Clarifying the extensive robotic grasping experiments**
> > >
> > > Thank you for finding our application of embodiment-transfer in robotic grasping tasks highly interesting. We are sorry for any confusion the original layout may have caused regarding the depth of our downstream application experiments. We will improve the presentation by restructuring the text to prominently feature the extensive results currently detailed in the appendix. Specifically, we will explicitly highlight our comprehensive comparisons against five baselines across three representative categories of dexterous grasp generation methods (analytical, generative, and transfer-based). Furthermore, we will bring more attention to our real-world humanoid robot deployments. We firmly believe that, as a downstream application, these extensive experiments are more than sufficient to demonstrate that our method can effectively facilitate dexterous manipulation tasks, ensuring that the practical value and effectiveness of our embodiment-transfer approach are clearly conveyed to the readers.
> > >
> > > Thank you again for your warm encouragement and for helping us strengthen our paper!

---

### Official Review · Reviewer_yy3P · 2026-03-08

**Soundness:** 3
**Presentation:** 3
**Significance:** 3
**Originality:** 3
**Overall Recommendation:** 5
**Confidence:** 4

**Summary:**

This paper addresses the generalization problem in monocular 3D shape reconstruction and proposes a shape deformation framework guided by **2D Foundation Models**. The core idea is to propagate 2D semantic features to the surface of the 3D template through a geometric guidance mechanism and utilize the view-adaptive module to address the deviation caused by perspective differences. The paper verified the practical value of this method through the robot grasping task.

**Compliance With Llm Reviewing Policy:**

Affirmed.

**Final Justification:**

To encourage the author, I recommend Accept (5). The authors' rebuttal effectively addressed my primary concerns regarding the evaluation landscape and computational efficiency.

The inclusion of Table R1 and Figure R1 significantly strengthens the paper by positioning the method against modern generative baselines (e.g., LRM, Phidias). These results demonstrate that the proposed deformation-based approach is not only substantially faster (under 2s vs. 4-120s) but also superior in preserving topological consistency for occluded regions.

While some presentation issues were noted by other reviewers, I consider the overall clarity sufficient and the robotic grasping application highly significant for proving practical utility. The authors' plan to integrate these new comparisons and the TAX3D baseline into the final version further solidifies the work's contribution. The strengths in originality (bridging 2D foundation models with 3D templates) and significance now clearly outweigh the initial weaknesses.

**Key Questions For Authors:**

Questions:

1. If there is a Genus difference between the target object and the template (for example, the template is a solid ball, but the target is a ring with holes), can the current Flow Matching architecture handle this inconsistency in topological structure?

2. When there is a significant conflict between the semantic features captured by the 2D basic model and the geometric features extracted by the 3D encoder, how does the model make weight trade-offs?

3. When evaluating the "unseen category," the experiment used the corresponding category template (such as reconstructing the cup with the cup template). If an unrelated template is randomly selected in a completely unknown category (such as reconstructing a cup with a car template), where is the failure threshold of the model?

**Limitations:**

yes

**Strengths And Weaknesses:**

Strengths:

1. The strong semantic representation of 2D basic models such as DINOv3 was successfully introduced into the 3D deformation task, significantly improving the reconstruction performance of the model on Unseen Categories.

2. The proposed geometric-guided Feature Propagation mechanism effectively solves the occlusion problem in a single viewing Angle and infers the semantic features of the backlight surface through geometric affinity.

3. The paper is not limited to geometric indicators. It also proves the reliability of the reconstruction results in complex downstream tasks (such as dexterous grasping) through Isaac Gym simulation and experiments with physical humanoid robots.

4. Experiments show that even under the **random templat** setting (i.e., when the template and the target shape differ greatly), this method can still maintain a high reconstruction accuracy.

Weaknesses:

1. Although the paper claims to be generalized, its actual operation still requires a 3D template of the same category as a starting point. When faced with a true "open world" or completely unknown strange objects, there is a lack of discussion in the text on how to obtain or generate an appropriate initial topological structure.

2. The feature propagation mechanism assumes that the semantics of geometrically adjacent or related regions are consistent. However, for objects with multiple materials or complex functional components, relying solely on geometric structures to disseminate semantic features may lead to distortion of the deformation field.

---

> ### Author Rebuttal · Authors · 2026-03-31
>
> Dear Reviewer yy3P, thanks for your time and valuable comments. We are glad you noted our method "successfully introduced" 2D models, "effectively solves the occlusion problem", "significantly improves the reconstruction performance", and "proves the reliability" in "complex downstream tasks". We have provided detailed responses to each of your questions below:
>
> ---
>
> **W1: Clarification on template acquisition to unknown objects**
>
> **A1**: We thank the reviewer for this thoughtful question. First, we would like to emphasize that as a shape deformation learning framework, our method is inherently robust to random and unseen templates, as demonstrated in Table 1 of the main paper. When presented with a novel object, the system first uses an open-vocabulary detector (e.g., Grounded-DINO) to identify its category from RGB data. If the object category is completely unknown, we can leverage generative text-to-mesh models (e.g., Meshy) to dynamically synthesize a shape template online. This generated template provides the necessary topological starting point for our model and is subsequently added to the library, allowing the system to continuously expand its capabilities over time. We will append these details in our revised manuscript.
>
> ---
>
> **W2 & Q2: Clarification on the geometry-guided feature propagation module**
>
> **A2**: Thanks for these questions. We agree that there are inherent discrepancies between 3D and 2D representations, which is precisely the motivation behind our geometry-guided modeling. Specifically, 3D features excel at capturing accurate local geometry and spatial topology, whereas 2D foundation features provide rich, high-level semantic understanding. Based on this, to ensure the reliability of the 2D representations, we follow common practice [R1] by utilizing DINO's deep patch features. As shown in [Figure R3](https://anonymous.4open.science/r/anony-k5a5X4t/FigureR3.png), these deep features encode high-level object semantics, providing stable representations that are more robust to appearance variations. Second, our geometry-guided module is designed to use 3D geometric priors to regularize the 2D semantic features for shape modeling. For objects with large shape differences but clear semantics (e.g., topologically diverse mug handles), the 2D features enhance the 3D representation to handle shape variations. Conversely, in regions where 3D geometry is continuous but 2D semantic features might be noisy, the 3D geometric features guide the propagation process to maintain structural plausibility. To enable the model to implicitly learn this dynamic trade-off, we employ extensive data augmentations during training, including variations in the template's materials, textures, and viewpoints. This encourages the propagation module to learn how to suppress unreliable 2D semantic noise under the guidance of 3D geometry. We will clarify these design choices in the revised manuscript.
>
> R1: DenseMatcher: Learning 3D Semantic Correspondence for Category-Level Manipulation from a Single Demo, ICLR, 2025.
>
> ---
>
> **Q1: Discussion on robustness to topological changes between the template and target**
>
> **A1**: We thank the reviewer for this question. Due to space limits, please refer to our response to **R1W3** for more details on how our model manages topological changes.
>
> ---
>
> **Q3: Model performance with templates from different categories**
>
> **A3**: We thank the reviewer for this question. To systematically investigate the model performance under cross-category template assignments, we grouped our test objects into three super-categories: 1) Furniture (chair, table, cabinet), 2) Container (bowl, bottle, mug, teapot, lotion pump), and 3) Vehicle (car, airplane). Without any retraining, we evaluated the model under three template selection settings:
>
> + Same-sub: Template from the exact same sub-category.
> + Same-super: Template from a different sub-category but within the same super-category.
> + Cross-super: Template from a different super-category.
>
> The quantitative and qualitative results are provided in [Figure R4](https://anonymous.4open.science/r/anony-k5a5X4t/FigureR4.png). As expected, when moving from *Same-sub* to *Same-super*, the CD and EMD errors increase; however, our method is still competitive with Phidias-3D (a template-augmented 3D generative model). When moving to *cross-super*, the errors increase substantially, marking the failure threshold of our model.
> These results illustrate how our method leverages 2D observations for guidance, while the template's topology acts as a structural regularizer during shape deformation. In same-super cases (e.g., bowl template to bottle target), the model matches the bottle's overall shape while preserving the bowl's open-top topology. Conversely, in cross-super cases, the extreme topological discrepancy exceeds the model's deformation capacity, causing a significant performance drop. We will include the above analysis in the revised manuscript.

---

> > ### Author Rebuttal · Reviewer_yy3P · 2026-04-02
> >
> > Thank you for your reply, which has solved my problem. I will keep my score.

---

> > > ### Author Response · Authors · 2026-04-05
> > >
> > > Dear reviewer yy3P, we sincerely thank you for your time and your continued endorsement of our paper. We will ensure that the clarifications provided during the rebuttal are integrated into the final version. Thank you again!

---

### Official Review · Reviewer_QGcP · 2026-03-10

**Soundness:** 2
**Presentation:** 3
**Significance:** 3
**Originality:** 3
**Overall Recommendation:** 4
**Confidence:** 4

**Summary:**

This paper introduces a generalizable framework for monocular 3D shape recovery via template deformation. To overcome the limitations of training visual encoders from scratch, the authors leverage pre-trained 2D foundation models (DINOv3) to extract robust semantic features. The method bridges the 2D-3D gap through a "geometry-guided feature propagation" module that diffuses visible 2D features across the entire 3D template. To handle viewpoint discrepancies between the template and the target, it employs a view-adaptive feature aggregation module that uses multi-view template renderings and camera pose embeddings. The actual deformation is modeled as a continuous trajectory using conditional Flow Matching. The authors demonstrate strong generalization to unseen categories and validate the practical utility of their dense correspondences by transferring contact maps for real-world robotic grasping.

**Compliance With Llm Reviewing Policy:**

Affirmed.

**Final Justification:**

My concerns are resolved, and I kept my positive rating of this paper.

**Key Questions For Authors:**

Please see weakness.

**Limitations:**

Please see weakness.

**Strengths And Weaknesses:**

Strengths:
1. The paper is written clearly. Figures are direct and informative and qualitative examples are clear.

2. The paper extends the 3D reconstruction to downstream real-world robotic manipulation. This proves dense correspondences generated by the deformation field can be used to successfully transfer human grasp contact maps to a physical humanoid robot. It provides practical value of this method.

3. The view-adaptive feature aggregation module is a principled solution to a common problem in template-based matching.

Weaknesses:
1. My biggest concern is the missing of comparison to recent feed-forward generative 3D methods (including LRM, shapelrm, wonder3D and so on). The author mainly compares current method with other deformation based baselines, and ignore another line of current state-of-the-art for single-view shape recovery. This makes it difficult to assess where this method truly stands in the broader 3D reconstruction landscape.

2. I am also curious about the efficiency. I am ok with additional computational overhead, but when compare to feed-forward generative methods, is this method faster or slower? The paper does not thoroughly discuss the computational overhead / time efficiency this method requires compared to standard single-view baselines.

3. I believe a fundamental limitation of continuous deformation is the inability to alter the inside/hole of an object, like adding or removing a hole, like a mug. I would recommend more discussion of these in its limitations, and how theoretical limitations impact some final results.

---

> ### Author Rebuttal · Authors · 2026-03-31
>
> Dear Reviewer QGcP, thanks for your time and constructive comments. We are glad you found our paper "written clearly" with "direct and informative" figures, and noted our method is a "principled solution to a common problem" and our robotic task "provides practical value." In the following, we respond to your questions in detail one by one:
>
> ---
>
> **W1: Additional comparison with 3D generative methods**
>
> **A1**: We thank the reviewer for this suggestion. To better position our work within the 3D reconstruction landscape, we compared our method against recent state-of-the-art 3D generative models, including LRM (ICLR’24), Wonder3D (CVPR’24), and Phidias (ICLR’25), across all test objects used in our main paper. As shown in Table R1 and [Figure R1](https://anonymous.4open.science/r/anony-k5a5X4t/FigureR1.png), while these baselines achieve reasonable shape recovery for visible regions, they struggle with self-occluded areas in single-view inputs, often producing shapes that are structurally incomplete. For example, in the case of the mug where the handle is partially occluded, the 3D generative methods only reconstruct the visible fragments, resulting in a broken, disconnected handle. Similarly, for the table whose legs are heavily obscured by the tabletop, most baselines fail to infer the hidden geometry, generating incomplete tables with missing legs. In contrast, our method explicitly deforms the template geometry, which preserves the underlying topology and leads to much more stable single-view reconstructions. We will incorporate these comparisons into the revised manuscript to better highlight our advantages.
>
> Table R1. Quantitative comparison with 3D generative methods for single-view shape recovery.
>
> | Method         | CD ($10^{-3}$) | EMD ($10^{-2}$) | S-IoU (%) | Time per shape (s) |
> | -------------- | -------------- | --------------- | --------- | ------------------ |
> | LRM-small      | 26.67          | 14.26           | 28.19     | 4.24               |
> | LRM-base       | 20.32          | 10.67           | 30.23     | 7.46               |
> | LRM-large      | 23.15          | 12.28           | 29.07     | 9.71               |
> | Wonder3D       | 19.31          | 10.52           | 29.92     | 121.57             |
> | Phidias-Image  | 9.52           | 7.97            | 33.06     | 5.44               |
> | Ours-random    | 3.09           | 5.14            | 49.19     | **0.60**           |
> | Ours-retrieval | **2.85**       | **4.95**        | **50.08** | 1.90               |
>
> ---
>
> **W2: Clarification on computational cost**
>
> **A2**: We thank the reviewer for this question. To clarify, our method is significantly faster than the feed-forward generative baselines. We have included the per-object inference time (measured on a single H800 GPU) in the last column of Table R1. While the generative baselines take anywhere from 4 to 120 seconds per shape, our approach leverages single-step flow matching to achieve high efficiency. Specifically, our method requires only **~0.6 seconds** per object with a randomly selected template, and **under 2 seconds** even when incorporating the template retrieval step. We will add a detailed discussion regarding the time efficiency to the revised manuscript.
>
> ---
>
> **W3: Discussion on robustness to topological changes between the template and target**
>
> **A3**: We appreciate the reviewer pointing out this point. We agree that altering the topology of an object is challenging for continuous deformation methods. To mitigate this, we leverage the 2D semantic similarity between the template and the target observation to guide the shape deformation, effectively relaxing strict geometric constraints. As shown in Figure 3 of the main paper, our method successfully manages several cases, such as morphing a four-legged table into a solid base, a two-tier table into a single-tier one, or adding armrests to a chair. To further validate this, we ablated the entire 2D feature modeling module and instead conditioned the model solely on the target features. Under the random template setting, the CD error more than doubled (3.09e-3 to 6.74e-3). The qualitative results in [Figure R2](https://anonymous.4open.science/r/anony-k5a5X4t/FigureR2.png) clearly show that without our 2D feature modeling, the model's robustness to large shape variations drops significantly, often failing to bridge topological gaps.
>
> Although our design pushes the boundaries of what continuous deformation fields can achieve compared to prior methods, we acknowledge that handling extreme topological changes remains constrained, as our point-level deformation relies on regularization to maintain local structural integrity. As shown in [Figure R4](https://anonymous.4open.science/r/anony-k5a5X4t/FigureR4.png), when the template and target belong to different categories with drastic shape discrepancies, the deformation results will be impacted. We will add this ablation and discussion to the revised manuscript.

---

> > ### Author Rebuttal · Reviewer_QGcP · 2026-04-03
> >
> > hanks to the authors for their response. I have read both the response and the comments from the other reviewers. My concerns are addressed, and I will remain my positive score.

---

> > > ### Author Response · Authors · 2026-04-05
> > >
> > > Dear reviewer QGcP, we appreciate your continued support and positive evaluation of our work. We will carefully incorporate the rebuttal discussions into the final manuscript to further improve the paper. Thank you again!

---

### Decision · Program_Chairs · 2026-04-30

**Decision:**

Accept (regular)

**Comment:**

This paper presents a strong and well-rounded contribution on monocular 3D shape recovery via template deformation. Its central technical contribution—geometry-guided modeling of 2D foundation-model features, combined with view-adaptive multi-view aggregation in a flow-matching deformation framework—is clear, well motivated, and meaningfully advances prior template-based reconstruction methods. The approach directly addresses robustness to large template-target shape variation, viewpoint change, and unseen categories.

The empirical results are strong: the method consistently outperforms prior deformation baselines across retrieved-template and random-template settings, including challenging unseen-category evaluations, and the qualitative examples show more plausible reconstruction under severe occlusion and structural variation. Another main strength of the paper are the downstream grasp-transfer results in simulation and on a real robot, which demonstrate that the learned correspondences are practically useful beyond reconstruction metrics alone.

The main reviewer concerns were about missing comparisons, efficiency, and limitations under topological mismatch and the rebuttal addressed these satisfactorily. Overall, the paper presents an excellent technical idea, strong results, and applications in robotics that could have significant impact.